# The molecular logic of Gtr1/2- and Pib2-dependent TORC1 regulation in budding yeast

**Jacob H Cecil[1], Cristina M Padilla[1], Austin A Lipinski[2], Paul Langlais[2], Xiangxia Luo[1], Andrew P Capaldi[1,3]***

[1]Department of Molecular and Cellular Biology, University of Arizona, Tucson, United States; [2]Department of Medicine, University of Arizona, Tucson, United States; [3]Bio5 Institute, University of Arizona, Tucson, United States

**\*For correspondence:**
capaldi@email.arizona.edu

**Competing interest:** The authors declare that no competing interests exist.

## eLife assessment

The study presents **valuable** findings concerning how a highly conserved signal transduction pathway helps budding yeast cells adapt their growth to nitrogen sources of differing qualities. However, the evidence is **incomplete** for the authors' main claim that the pathway adopts three distinct states depending on the nitrogen source. The presented data, particularly phospho-proteomic datasets, will be of interest to the cell growth signaling community.

**Abstract** The Target of Rapamycin kinase Complex 1 (TORC1) regulates cell growth and metabolism in eukaryotes. Previous studies have shown that, in *Saccharomyces cerevisiae*, nitrogen and amino acid signals activate TORC1 via the highly conserved small GTPases, Gtr1/2, and the phosphatidylinositol 3-phosphate binding protein, Pib2. However, it was unclear if/how Gtr1/2 and Pib2 cooperate to control TORC1. Here, we report that this dual regulator system pushes TORC1 into at least three distinct signaling states: (i) a Gtr1/2 on, Pib2 on, rapid growth state in nutrient replete conditions; (ii) a Gtr1/2 inhibited, Pib2 on, adaptive/slow growth state in poor-quality growth medium; and (iii) a Gtr1/2 off, Pib2 off, quiescent state in starvation conditions. We suggest that other signaling pathways work in a similar way to drive a multilevel response via a single kinase, but the behavior has been overlooked since most studies follow signaling to a single reporter protein.

## Introduction

The Target of Rapamycin kinase Complex 1 (TORC1) is the master regulator of cell growth and metabolism in eukaryotes (*Loewith and Hall, 2011*; *Liu and Sabatini, 2020*; *González and Hall, 2017*). In the presence of pro-growth hormones and abundant nutrients, TORC1 is active and drives growth by stimulating protein, ribosome, lipid, and nucleotide synthesis (*Loewith and Hall, 2011*; *Liu and Sabatini, 2020*; *González and Hall, 2017*; *Robitaille et al., 2013*; *Peterson et al., 2011*; *Huber et al., 2009*; *Hsu et al., 2011*; *Ben-Sahra and Manning, 2017*; *Ben-Sahra et al., 2016*). In contrast, when hormone or nutrient levels drop, TORC1 is inhibited, causing the cell to switch from anabolic to catabolic metabolism and eventually enter a quiescent state (*Kim et al., 2011*; *Kamada et al., 2000*; *Düvel et al., 2010*).

Numerous proteins and pathways have been shown to regulate TORC1, including (i) the small GTPases Rag A/B and C/D (*Sancak et al., 2008*; *Sancak et al., 2010*; *Kim et al., 2008*; *Efeyan et al., 2013*; *Bar-Peled et al., 2012*; *Binda et al., 2009*; *Dubouloz et al., 2005*) and the associated GTPase Activation Protein (GAP), GATOR1/2 (*Bar-Peled et al., 2013*); (ii) the small GTPase Rheb and

its associated GAP, the Tuberous Sclerosis Complex (*Yang et al., 2017*; *Menon et al., 2014*; *Dibble and Manning, 2013*); (iii) the AMP-activated protein kinase (*Inoki et al., 2003b*; *Inoki et al., 2003a*; *Gwinn et al., 2008*); (iv) the Nemo-like kinase (*Yuan et al., 2015*); (v) the cAMP-dependent protein kinase (*Jewell et al., 2019*); and (vi) the small GTPase ADP-ribosylation factor 1 (*Meng et al., 2021*; *Li et al., 2010*). However, it remains unclear how the proteins/pathways listed above (and others) cooperate to control TORC1.

Here, to address this question, we examine signal integration at TORC1 in the simple model organism, *Saccharomyces cerevisiae*.

Previous work in *S. cerevisiae* has shown that:

1. Amino acid and nitrogen signals are transmitted to TORC1 via a heterodimeric pair of small GTPases that are homologous to RagA/B and RagC/D, called Gtr1 and Gtr2 (*Binda et al., 2009*; *Dubouloz et al., 2005*). Specifically, when cells are grown in medium containing a high concentration of amino acids (or a high-quality nitrogen source), Gtr1/2 are in their GTP and GDP-bound forms, respectively, and bind to/activate TORC1 (*Binda et al., 2009*; *Dubouloz et al., 2005*). However, once amino acid/nitrogen levels fall, SEAC (a homolog of GATOR1/2) triggers GTP hydrolysis at Gtr1 to drive the complex into the Gtr1-GDP, Gtr2-GTP-bound state, and inhibit TORC1 (*Panchaud et al., 2013*; *Neklesa and Davis, 2009*; *Laxman et al., 2014*; *Chen et al., 2017*; *Algret et al., 2014*).

2. TORC1 is also regulated by the phosphatidylinositol 3-phosphate binding protein, Pib2 (*Hatakeyama, 2021*; *Kim and Cunningham, 2015*). Much less is known about Pib2 than Gtr1/2, but several facts are clear: first, Pib2 binds directly to TORC1 and activates the complex via its highly conserved C-terminal domain (CAD) domain (*Kim and Cunningham, 2015*; *Troutman et al., 2022*; *Tarassov et al., 2008*; *Tanigawa and Maeda, 2017*; *Michel et al., 2017*). Second, Pib2 can repress TORC1 via its N-terminal inhibitory domain (NID) (*Michel et al., 2017*). Third, Pib2 activates TORC1 in response to glutamine (*Tanigawa and Maeda, 2017*; *Ukai et al., 2018*; *Tanigawa et al., 2021*).

Most data but how and why do Gtr1/2 and Pib2 work together to regulate TORC1?

Most data suggest that Gtr1/2 and Pib2 act in parallel (redundantly) to activate TORC1 (*Hatakeyama, 2021*). For example, yeast missing either Gtr1/2 or Pib2 grow well in nutrient-rich media, while *gtr1/2Δpib2Δ* cells are sick/dead (a phenotype that can be rescued by a hyperactive TOR allele) (*Kim and Cunningham, 2015*). Furthermore, *gtr1/2Δ* and *pib2Δ* cells have strong TORC1 activity as measured by the downstream reporters phospho-Rps6 and Sch9, while transient repression of Pib2 in a *gtr1/2Δ* cell line blocks TORC1 signaling to these same proteins (*Kim and Cunningham, 2015*; *Ukai et al., 2018*).

However, other data suggest that the impact that Gtr1/2 and Pib2 have on TORC1 signaling is more complex: First, *gtr1Δ*, *gtr2Δ*, *gtr1/2Δ*, and *pib2Δ* cells are all hypersensitive to the TORC1 inhibitor rapamycin (*Michel et al., 2017*). Second, both *gtr1/2Δ* and *pib2Δ* cells fail to activate TORC1 when leucine or glutamine are added back to cells treated with rapamycin (*Varlakhanova et al., 2017*). Third, phosphorylation of the TORC1 target Npr1, and its substrate Par32, is sensitive to deletion of either Pib2 or Gtr1/2 (*Kim and Cunningham, 2015*; *Brito et al., 2019*).

In this report, we build on the previous studies by examining the impact that Gtr1/2 and Pib2 have on TORC1 signaling across the proteome using a combination of phosphoproteomics and standard reporter assays. The resulting data show that Gtr1/2 and Pib2 are both required for full TORC1 activation. Importantly, however, deletion of Gtr1/2 or Pib2 only blocks signaling to a subset of the TORC1 substrates—primarily those involved in amino acid metabolism and nutrient transport. These observations lead us to propose a new model where partial starvation triggers metabolic reprogramming via TORC1 (by inactivating Gtr1/2 or Pib2) but does not block cell growth. And in follow-up experiments, we confirm that this is, indeed, the case. Specifically, we show that when yeast are first transferred from medium containing a high-quality nitrogen source, to medium containing a low-quality nitrogen source, TORC1 is completely inhibited to block growth and activate metabolic reprogramming. Then, as the cells adapt to the low-quality nitrogen source, Pib2 is turned on again to reinitiate growth, while Gtr1/2 remains inhibited, or partially inhibited, to ensure that the cells continue to activate the metabolic pathways and transporters necessary for adaptation/survival.

Thus, the TORC1 circuit in yeast uses two different amino acid/nitrogen signaling proteins to drive the cell into a rapid growth state, an adaptive growth state, or a quiescent state, depending on the

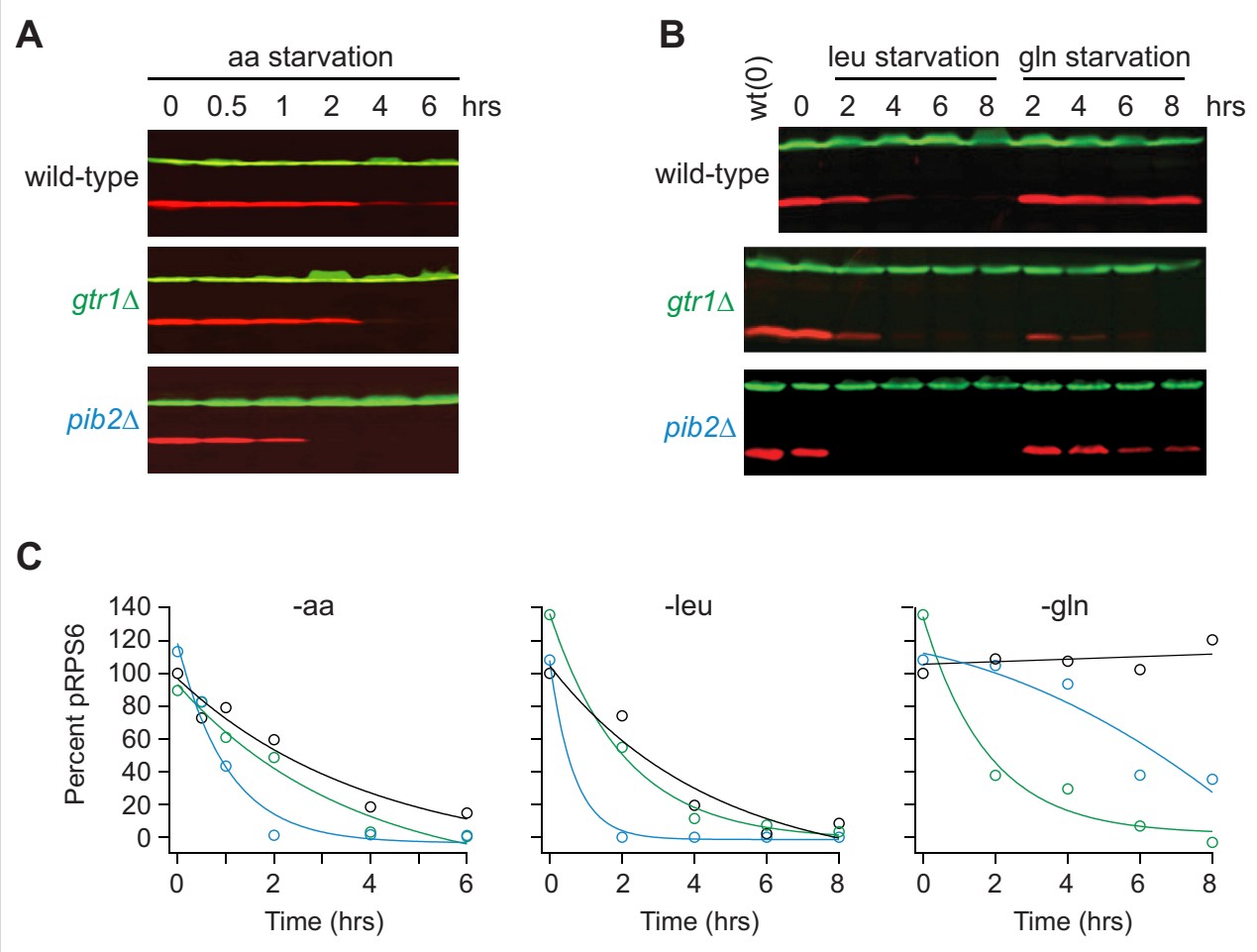

**Figure 1.** Impact of Gtr1/2 and Pib2 on TORC1 signaling during amino acid starvation. (**A**) TORC1 activity measured before and after complete amino acid starvation in wild-type, *gtr1Δ*, and *pib2Δ* strains using a western blot with anti phospho-Rps6 (red) and anti-PGK (green) antibodies. (**B**) TORC1 activity measured as in (**A**) but during leucine and glutamine starvation. Glutamine starvation was triggered by transferring the cells from synthetic complete (SC) medium, to SC medium missing glutamine and containing 2 mM methionine sulfoximine (MSX; a glutamine synthetase inhibitor). Leucine starvation was triggered by transferring the three leu⁻ strains from SC medium to SC medium missing leucine. (**C**) Values showing the ratio of the p-Rps6 signal divided by the PGK (loading control) signal in each lane from (**A**) and (**B**), relative to the value for the wild-type strain at time = 0.

The online version of this article includes the following source data for figure 1:

**Source data 1.** Original western blots for *Figure 1A and B* indicating the relevant bands and treatments.

**Source data 2.** Original files for western blots displayed in *Figure 1*.

environmental conditions. We argue that other signaling pathways probably work in a similar way, but multilevel responses are overlooked since most studies follow signaling to a single reporter.

## Results

### Reporter (pRps6)-based analysis of Gtr1/2- and Pib2-dependent signaling

Gtr1/2 and Pib2 are thought to transmit leucine, glutamine, and other amino acid signals to TORC1 (*Hatakeyama, 2021*). To test this model and learn more about the cooperation between Gtr1/2 and Pib2, we followed the phosphorylation of a downstream reporter of TORC1 activity (Rps6) in wild-type, *gtr1Δ*, and *pib2Δ* cells during amino acid starvation (*Yerlikaya et al., 2016*; *Chen et al., 2018*). These experiments showed that all three strains have the same TORC1 activity in nutrient-replete medium (time 0, *Figure 1*), consistent with the idea that Gtr1/2 and Pib2 work in parallel (redundantly) to activate TORC1. These experiments also showed that TORC1 turns off efficiently in the absence

of Gtr1 or Pib2 during leucine and complete amino acid starvation (*Figure 1*). However, the results in glutamine starvation stood out; TORC1 remained active in the wild-type strain, and partially active in the *pib2Δ* strain, but was rapidly inhibited in the *gtr1Δ* strain (*Figure 1*).

To learn more about the interaction between Gtr1/2 and Pib2 during glutamine starvation, we followed Rps6 phosphorylation in cells missing the N-terminal Inhibitory Domain of Pib2 (*pib2NIDΔ*). The *pib2NIDΔ* mutation completely blocked TORC1 inhibition in the *gtr1Δ* background but had no impact on TORC1 signaling in a wild-type background (*Figure 2A and B*), demonstrating that Pib2 helps drive TORC1 inactivation during glutamine starvation (at least in the absence of Gtr1).

We then measured Rps6 phosphorylation in a strain missing both Gtr1 and Gtr2 (*gtr1Δgtr2Δ*), and a strain carrying mutations that lock Gtr1 and 2 into their inactive Gtr1-GDP and Gtr2-GTP-bound forms (Gtr1$^{S20L}$ and Gtr2$^{Q66L}$; Gtr1/2$^{off}$ for short *Binda et al., 2009*). To our surprise, TORC1 remained active in both strains (*Figure 2A and B*), demonstrating that Pib2 can only inhibit TORC1 during glutamine starvation in strains where the Gtr1/2 complex is partially disrupted (*gtr1Δ* and *gtr2Δ* strains) but not completely absent or inactive (*Figure 2A and B*).

We also saw the same pattern when we examined cell survival in glutamine starvation. Wild-type, *pib2Δ*, *pib2NIDΔ*, *pib2NIDΔgtr1Δ*, Gtr1/2$^{off}$, and the other cell lines that failed to inactivate TORC1, all die at a higher rate during glutamine starvation than the *gtr1Δ* and *gtr2Δ* cells—presumably because they do not arrest cell growth in a timely manner (*Figure 2C*).

In sum, most of our measurements of Rps6 signaling and cell death fit with the prevailing model of TORC1 regulation, where Gtr1/2 and Pib2 (i) act in parallel (redundantly) to activate TORC1 in nutrient-replete conditions, and (ii) are switched off during amino acid starvation to inhibit TORC1. However, we were left with a puzzle. Our data also revealed that Pib2 can transmit glutamine starvation signals to TORC1, but those signals have little to no impact on TORC1 activity in the presence of an intact Gtr1/2 complex. Why would this be? We hypothesized that signals transmitted through Pib2 alone, or Gtr1/2 alone, do impact TORC1 signaling in a wild-type background, but the response is just not apparent at the level of Rps6 phosphorylation.

## Proteome-wide analysis of Gtr1/2- and Pib2-dependent signaling

To learn more about Gtr1/2 and Pib2 signaling, we used mass spectrometry to quantify the global protein phosphorylation levels in wild-type, *gtr1Δgtr2Δ*, and *pib2Δ* cells grown in nutrient-replete medium (SC), and in wild-type cells treated with the TORC1 inhibitor rapamycin (all in quadruplicate). In the end, we were able to quantify the level of 7807 phosphopeptides (covering 5325 phosphosites on 1686 proteins) across the 16 samples (*Supplementary file 1*). 445 of these phosphopeptides (covering 362 phosphosites on 301 proteins) were up- or downregulated in response to rapamycin treatment, deletion of Gtr1/2, and/or the deletion of Pib2 (more than twofold change and p<0.01 in one or more strain/condition). More specifically, 175 phosphopeptides were downregulated in rapamycin (*Figures 3 and 4*), 187 phosphopeptides were upregulated in rapamycin (*Figure 4—figure supplement 1*), and 83 phosphopeptides were up- or downregulated in the *gtr1Δgtr2Δ* or *pib2Δ* cells, but not in the rapamycin-treated cells (*Figure 4—figure supplement 2*).

We focused our analysis on the 175 phosphopeptides that are downregulated in rapamycin since they cover most of the well-known targets of TORC1 and its downstream effectors, including Sch9, Tod6, Maf1, Stb3, Ypk3, Atg13, Mks1, Nnk1, Npr1, Par32, Avt1, Avt4, Sky1, Gat1, Gln3, Ume6, Rtg3, Lst4, Gcn2, Tco89, Ssd1, and Stp1 (*Figures 3 and 4*; *González and Hall, 2017*; *Hughes Hallett et al., 2014*; *Soulard et al., 2010*; *Shin et al., 2009*; *Dokládal et al., 2021*; *Cherkasova and Hinnebusch, 2003*; *Kamada et al., 2010*; *Urban et al., 2007*; *Huber et al., 2011*). This dataset revealed two novel aspects of Gtr1/2 and Pib2 signaling. First, TORC1 drives the phosphorylation of several residues near the N-termini of Ser3 and Ser33 (*Figures 3 and 4*), homologous 3-phosphoglycerate dehydrogenases that catalyze the first step in serine and glycine synthesis (*Albers et al., 2003*). Remarkably, these phosphorylation reactions are completely dependent on Pib2, but are not altered in the *gtr1Δgtr2Δ* strain (*Figures 3 and 4*). Second, many TORC1-dependent phosphorylation events (outside of Ser3/33) depend heavily on *both* Pib2 and Gtr1/2 (green labels, *Figure 3*, and left columns, *Figure 4*, *Figure 4—figure supplement 1*). However, other phosphorylation events are unperturbed by the deletion of Gtr1/2 or Pib2 (blue labels, *Figure 4*, *Figure 4—figure supplement 1*). In fact, there is a strong correlation between the impact that Pib2 and Gtr1/2 have on the TORC1-dependent phosphorylation (R=0.70 excluding Ser3/33; *Figure 4*), but that impact runs the gamut from matching

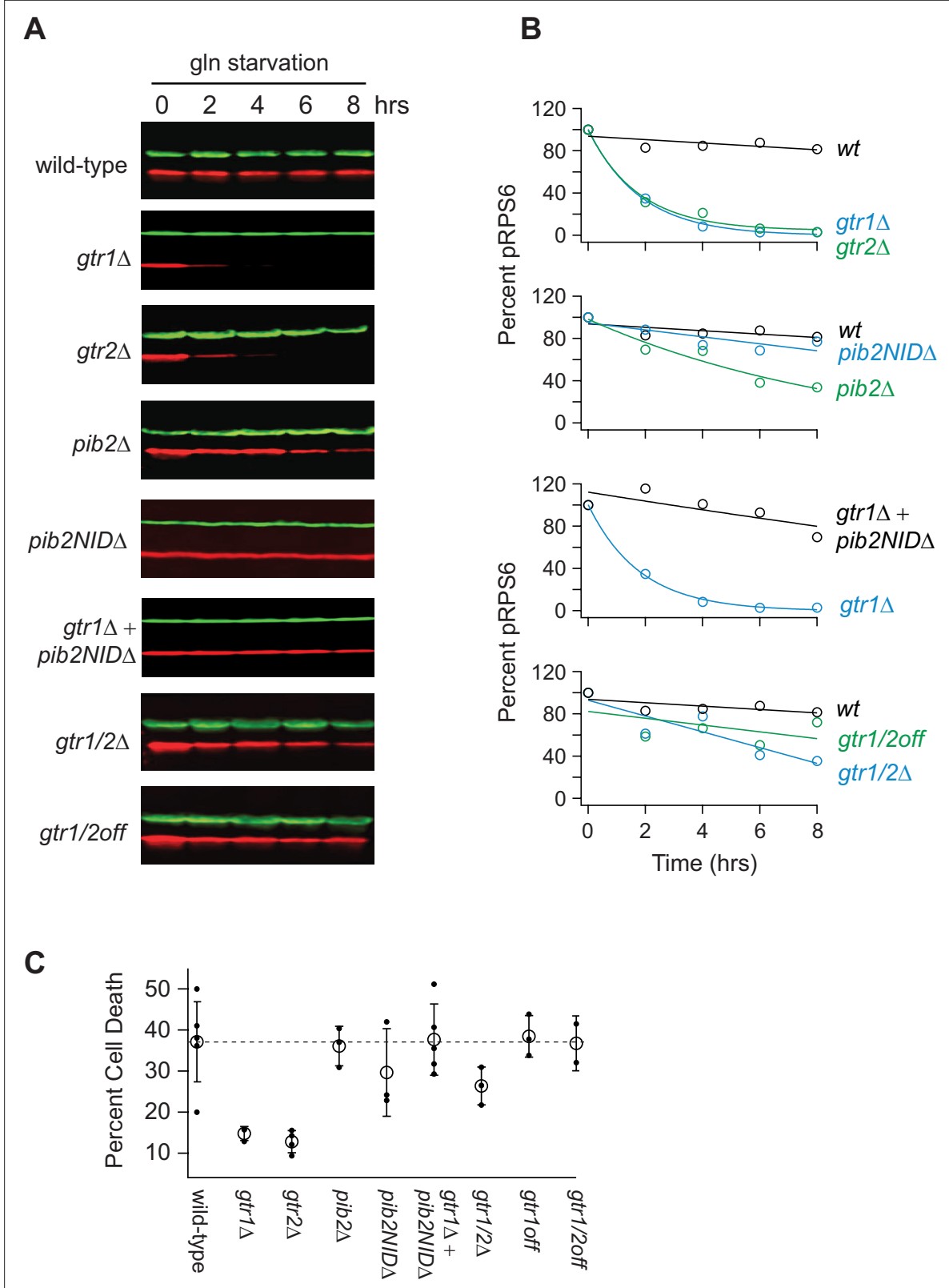

**Figure 2.** Impact of Gtr1/2 and Pib2 on TORC1 signaling during glutamine starvation. (**A**) TORC1 activity measured during glutamine starvation in *gtr1Δ*, *gtr2Δ*, *pib2Δ*, *pib2NIDΔ*, *gtr1Δpib2NIDΔ*, *gtr1Δgtr2Δ*, and *gtr1/2*[off] strains using a western blot, as described in *Figure 1*. (**B**) Values showing the ratio of the p-Rps6 signal divided by the PGK (loading control) signal in each lane from (**A**) relative to the value for the wild-type strain at time = 0. (**C**) Fraction of cells that are dead after 6 h of glutamine starvation for each strain listed in (**B**) as measured using SYTOX green labeling and a fluorescence microscope.

*Figure 2 continued on next page*

*Figure 2 continued*

The open circles and error bars show the average and standard deviation from four replicates (filled circles), with >200 cells analyzed per replicate, per strain.

The online version of this article includes the following source data for figure 2:

**Source data 1.** Original western blots for *Figure 2A*, indicating the relevant bands and treatments.

**Source data 2.** Original files for western blots displayed in *Figure 2*.

the impact of rapamycin, to zero (top left, to bottom right, continuum; *Figure 4*, *Figure 4—figure supplement 1*).

Examining the list of proteins that are heavily dependent on Pib2 and Gtr1/2, we noticed an obvious trend; 8/11 of the proteins with the strongest reliance on Pib2 and Gtr1/2 are involved in nutrient transport and utilization (top-left column, *Figure 4*). These proteins include the amino acid transporters Bap2, Agp1, Gnp1, and Hip1 (*Grauslund et al., 1995*; *Schreve et al., 1998*; *Tanaka and Fink, 1985*; *Zhu et al., 1996*); the zinc transporter Zrt3 and the zinc-regulated transcription factor, Zap1 (*MacDiarmid et al., 2000*; *Zhao and Eide, 1997*); the hexose transporter Hxt1 (*Lewis and Bisson, 1991*); and the TORC1-dependent regulator of amino acid transporter activity, Par32 (*Boeckstaens et al., 2015*). This trend was also clear when we carried out GO analysis on the full list of Gtr1/2- and Pib2-dependent proteins (left column, *Figure 4*), which includes the key TORC1-dependent regulator of amino acid metabolism, Npr1 (*Boeckstaens et al., 2015*; *MacGurn et al., 2011*; *Schmidt et al., 1998*), and the amino acid transporter Tat1 (*Schmidt et al., 1994*) (amino acid and aromatic amino acid transporter activity were the top hits; p=1e-4).

In contrast, the proteins that are dependent on TORC1, but not Gtr1/2 or Pib2 (right column, *Figure 4*), tend to be involved in cell signaling (p=1e-4) and the regulation of cell growth (biosynthetic processes, p=9e-3). This group of proteins includes many of the key regulators of ribosome and protein synthesis, including Sch9, Maf1, Stb3, Gcn2, and Kcs1 (*Huber et al., 2009*; *Dokládal et al., 2021*; *Cherkasova and Hinnebusch, 2003*; *Huber et al., 2011*; *Worley et al., 2013*), as well as the stress response factor Hsf1 (*Sorger and Pelham, 1988*) and the autophagy regulator Ksp1 (*Chang and Huh, 2018*; *Figure 3* and right column, *Figure 4*).

Thus, the view of Gtr1/2 and Pib2 signaling that we and others built up by following Rps6 phosphorylation is misleading (*Figures 1 and 2*). Gtr1/2 and Pib2 do not act redundantly during steady-state growth, but instead, are both required for full TORC1 activation. It is just that some TORC1 pathway targets (like Rps6) are efficiently phosphorylated even when TORC1 is partially active.

## TORC1- and Pib2-dependent regulation of Ser33

To gain insight into the function of, and mechanism underlying, TORC1- and Pib2-dependent signaling to Ser33 (the dominant enzyme in the Ser3/33 pair; *Paczia et al., 2019*), we first set out to map the TORC1- dependent phosphorylation sites on Ser33. Our global phosphoproteomics experiments had already identified serine 20, 22, 28, and 29 as TORC1- and Pib2-dependent sites (*Figures 3 and 4*). However, to see if we missed any sites due to under-sampling, we also immunopurified Ser33-FLAG from cells grown in SD medium ±rapamycin and mapped the phosphorylation sites using mass spectrometry. These experiments (and previously published phosphoproteomics data; *Dokládal et al., 2021*) indicated that serine 27, serine 33, and threonine 31 are also TORC1-dependent sites (*Figure 5A*; *Supplementary file 2*).

Next, we wanted to identify the stimuli that impact TORC1 signaling to Ser33. Using Phostag gel electrophoresis (*Kinoshita et al., 2005*), we found that rapamycin, glucose starvation, and complete nitrogen starvation (but not amino acid starvation) all lead to the rapid dephosphorylation of Ser33 (*Figure 5B*). As expected, mutation of the seven TORC1-dependent phosphorylation sites listed above to alanine, or deletion of Pib2, also blocked the rapamycin-dependent phosphorylation events detected on the gel (*Figure 5C and D*).

The pattern of regulation seen for Ser33—with strong glucose, rapamycin, and nitrogen, but limited amino acid, dependence—matched that seen previously for Sch9/Rps6 (*Hughes Hallett et al., 2014*; *Wallace et al., 2022*), leading to the question, what, if any, unique property (or properties) does TORC1-Pib2-dependent signaling to Ser33 have? To address this question, we first measured Ser33 phosphorylation during leucine and glutamine starvation. Here, we saw slow dephosphorylation

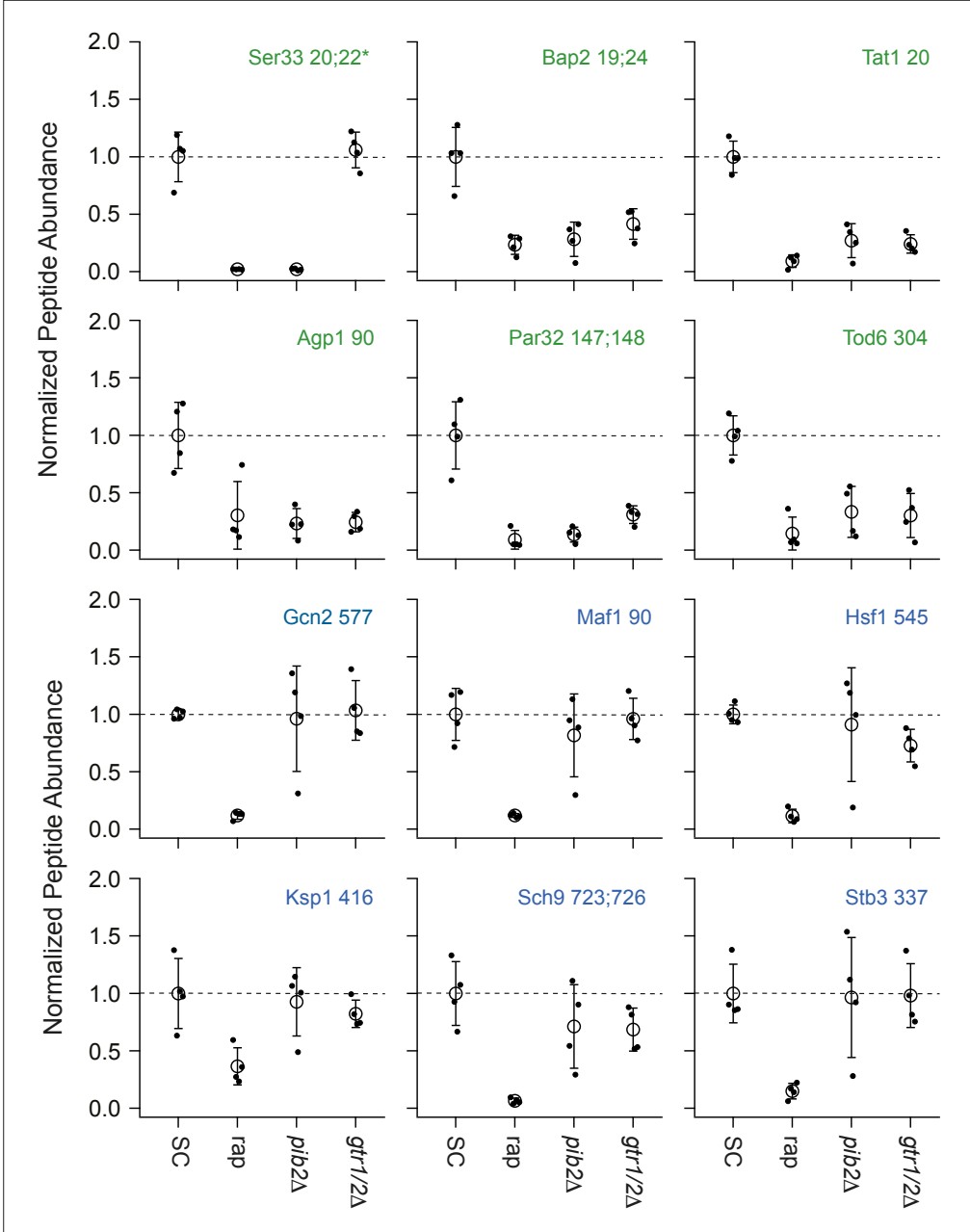

**Figure 3.** Impact of rapamycin, Gtr1/2 deletion, and Pib2 deletion on select TORC1-dependent phosphopeptides. Normalized peptide abundance for select phosphopeptides extracted from wild-type cells growing in synthetic complete (SC) medium, wild-type cells growing in SC medium and treated with 200 nM rapamycin for 30 min, or *pib2Δ* or *gtr1Δgtr2Δ* cells growing in SC medium (as labeled). The open circles and error bars show the average and standard deviation from four replicates (filled circles), and all data is divided by the average signal (for the relevant peptide) in the wild-type strain growing in SC medium. The phosphopeptides are named based on the protein they are from, followed by the number of the phosphorylated residue(s). Entries with an asterisk (*) indicate that there are other possible phosphorylation site assignments (see ***Supplementary file 1***). Phosphopeptides labeled in green (top panels) change significantly in the *pib2Δ* and/or *gtr1Δgtr2Δ* backgrounds.

in leucine starvation, and no change in glutamine starvation, just as with Rps6 (compare ***Figure 1*** and ***Figure 5E***). We then wondered if TORC1-Pib2 signaling to Ser33 responds to the quality of the nitrogen source in the growth medium. To test this, we followed Ser33 phosphorylation in a prototrophic strain as it transitioned from growth in a poor-quality nitrogen source (proline medium) to a high-quality nitrogen source (glutamine medium) (***Stracka et al., 2014***). This experiment revealed

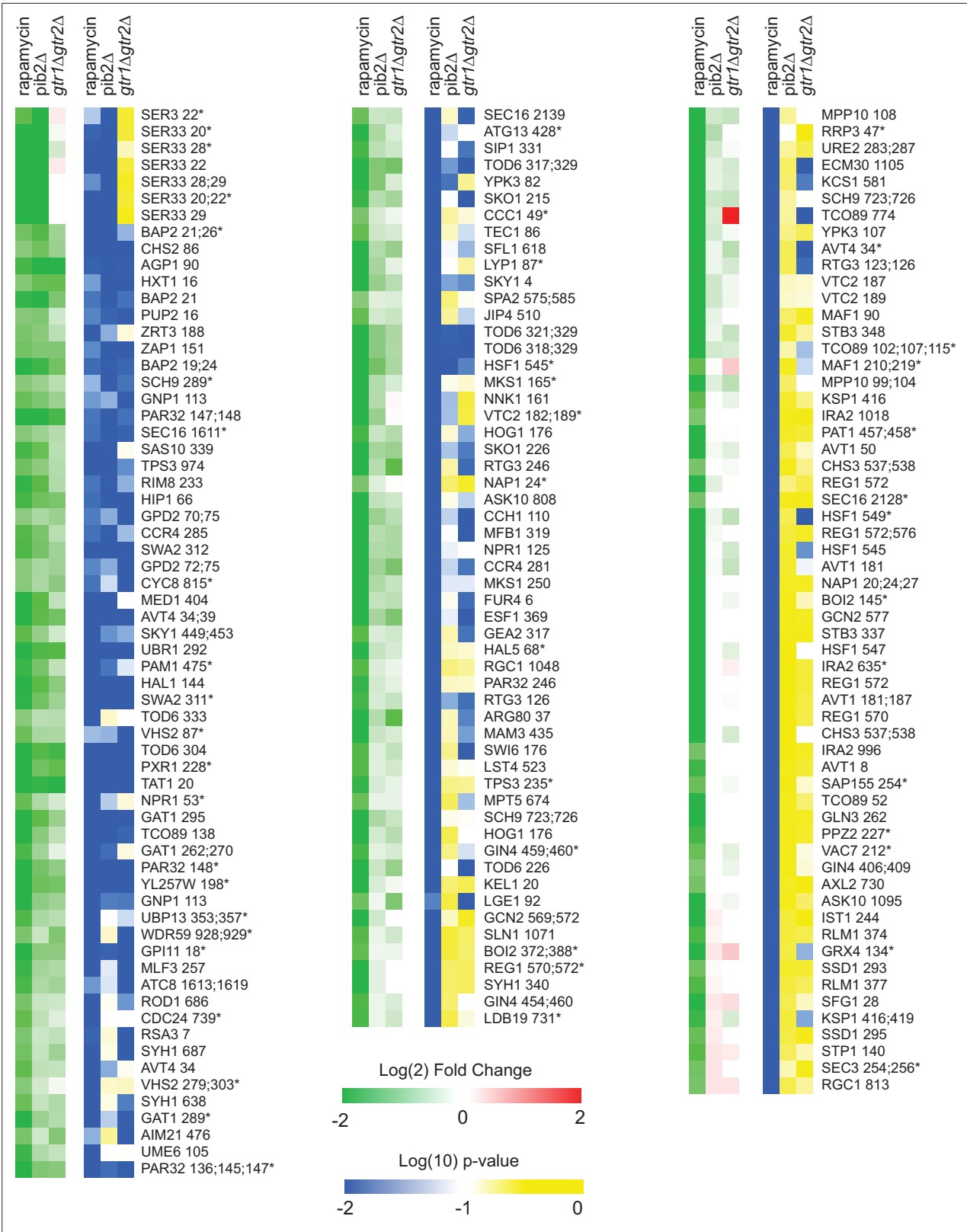

**Figure 4.** Phosphopeptides significantly downregulated in rapamycin. Heatmap showing the abundance of all the phosphopeptides that are downregulated in rapamycin more than twofold with p-value <0.01; t-test. Columns 1–3 (green-red) show the peptide levels after 30-min rapamycin treatment, in the *pib2Δ* strain growing in synthetic complete (SC) medium, and in the *gtr1Δgtr2Δ* strain growing in SC medium (as labeled), all compared to those in the wild-type strain growing in SC medium. The values are the average from four replicate experiments. Columns 4–6 (blue-yellow) show the

*Figure 4 continued on next page*

*Figure 4 continued*

statistical significance of any change in columns 1–3 based on a *t*-test. The expression data is ordered based on the fraction of the rapamycin response found in the *pib2Δ* strain (top left, to bottom right). The phosphopeptides are named based on the protein they are from, followed by the number of the phosphorylated residue(s). Entries with an asterisk (*) indicate that there are other possible phosphorylation site assignments (see *Supplementary file 1*).

The online version of this article includes the following figure supplement(s) for figure 4:

**Figure supplement 1.** Phosphopeptides significantly upregulated in rapamycin.

**Figure supplement 2.** Phosphopeptides significantly up/downregulated by deletion of Gtr1/2 or Pib2 but not by rapamycin.

that Ser33 phosphorylation is exquisitely sensitive to the quality of the amino acid/nitrogen source, with the TORC1-dependent sites going from unphosphorylated in proline, to >80% phosphorylated in glutamine (*Figure 5F*, *Figure 5—figure supplement 1A and B*). Other substrates we looked at (as detailed in *Figure 6*) do not have such a strong response during the proline to glutamine shift, suggesting that the Pib2 dependence underlies this unique connection between TORC1 and Ser33.

We also wanted to understand why the TORC1-dependent phosphorylation of Ser33 (and Ser3) requires Pib2. To address this question, we examined our recently published interactome data for Kog1 (the major regulatory subunit in TORC1) and Pib2 (*Wallace et al., 2022*). This analysis revealed that Ser3 and Ser33 are two out of a total of six proteins that are captured at a significantly higher level in a Pib2 immunopurification than in a Kog1 immunopurification (out of >200 proteins enriched in the Kog1 IP), suggesting that Ser3 and Ser33 bind to Pib2 and not TORC1 (*Figure 5—figure supplement 1D*). In line with this, we found that Ser33-HA is enriched in a Pib2 purification (2.5-fold) but not a Gtr1/2 purification (*Figure 5G*).

Finally, to see if the TORC1- and Pib2-dependent phosphorylation of Ser33 is important for cell function, we followed the growth of phosphonull (S7A) and phosphomimic (S7D) versions of Ser33 in medium missing serine and glycine (in a Ser3 delete background). These experiments showed that the phosphonull (but not the phosphomimic) strain has a significant delay exiting quiescence, but then grows at the same rate as the wild-type strain once it is dividing rapidly (*Figure 5H*). These data suggest (but do not prove) that the TORC1- and Pib2-dependent phosphorylation of Ser33 helps drive serine and glycine synthesis as cells transition into a rapid growth state, but TORC1-dependent phosphorylation is not required to maintain basal Ser33 activity during log phase growth.

## Multilevel signaling through TORC1

Our discovery that the deletion of Gtr1/2 or Pib2 leads to a change in TORC1 signaling through some substrates (particularly those involved in amino acid transport and metabolism), but not those involved in cell growth control, led us to hypothesize that the TORC1 pathway can take up at least three distinct signaling states (the first two of which are well known; *Figure 6A*): (I) a fully active state to promote cell growth and inhibit the Npr1-dependent amino acid starvation response. (II) An inactive state to block cell growth and activate the Npr1-dependent amino acid starvation response. (III) A partially active state to simultaneously promote cell growth and activate the Npr1-dependent amino acid starvation response.

To test this model (i.e., see if State III exists in a wild-type strain), we needed reliable reporters of the TORC1-dependent cell growth and amino acid starvation responses. We therefore turned to two well-established assays.

First, to follow the cell growth response, we monitored the phosphorylation of cleaved Sch9 using an SDS-PAGE mobility assay (*Hughes Hallett et al., 2014*; *Urban et al., 2007*). This assay follows the phosphorylation of several TORC1 target sites at the C-terminus of Sch9, known to play a key role in activating the kinase and protein synthesis. These C-terminal sites remain phosphorylated in the Gtr1/2 and Pib2 delete strains (Ser 723 and 726; *Figure 3* and right column *Figure 4*).

Second, to follow the TORC1-dependent amino acid starvation response, we monitored the phosphorylation of full-length Par32 using an SDS-PAGE mobility assay (*Hughes Hallett et al., 2014*). Par32 regulates amino acid transporter activity and is a substrate/reporter for the key TORC1-dependent amino acid starvation response regulator, Npr1 (*Boeckstaens et al., 2015*). Importantly, both Par32 and Npr1 are highly sensitive to the deletion of Gtr1/2 or Pib2 (left columns; *Figure 4*, *Figure 4—figure supplement 1*). In the case of Par32, some sites are dephosphorylated in rapamycin and the

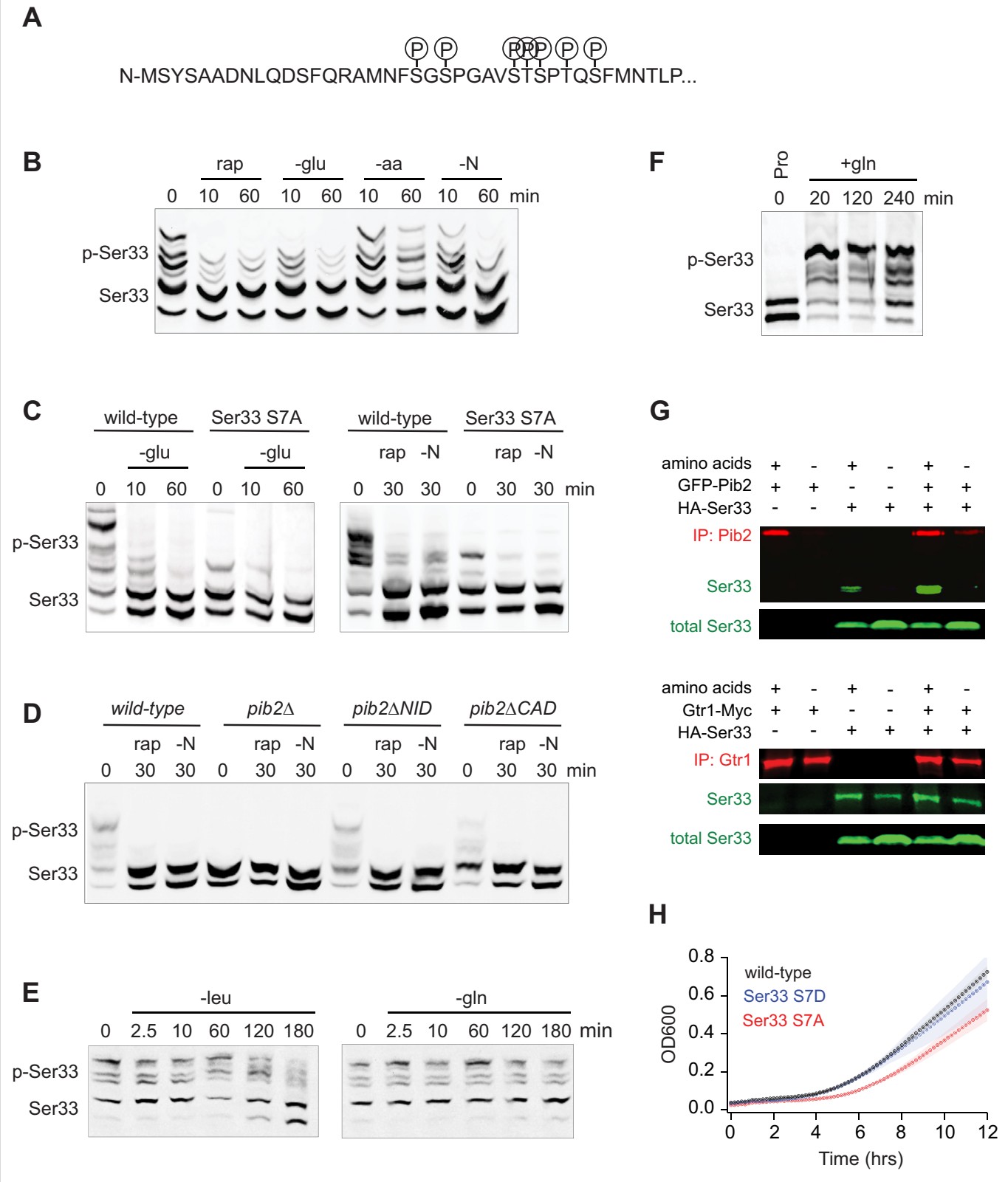

**Figure 5.** Function of, and mechanism underlying, the TORC1- and Pib2-dependent phosphorylation of the phosphoglycerate dehydrogenase Ser33. (**A**) TORC1-dependent phosphorylation sites compiled from global phosphoproteomics data, and mass-spectrometry-based analysis of Ser33 immunopurified from cells in mid-log phase before and after treatment with 200 nM rapamycin. (**B**) Phos-tag gel measuring Ser33 phosphorylation before and after treatment with 200 nM rapamycin (rap) or starvation for glucose (-glu), amino acids (-aa), or all nitrogen (-N). (**C**) Phos-tag gel comparing

*Figure 5 continued on next page*

*Figure 5 continued*

Ser33 phosphorylation in wild-type and Ser33[S7A] cells following glucose starvation (transfer to synthetic medium +3% lactate), rapamycin treatment, or nitrogen starvation. (**D**) Phos-tag gel examining Ser33 phosphorylation in wild-type, *pib2Δ*, *pib2ΔNID*, or *pib2ΔCAD* cells during log phase growth (0), after 30 min of rapamycin treatment (rap), or complete nitrogen starvation (-N). (**E**) Phos-tag gel examining Ser33 phosphorylation in cells grown to mid-log phase and then starved for leucine (left gel) or starved for glutamine and treated with MSX (right gel). (**F**) Phos-tag gel examining Ser33 phosphorylation in cells grown to mid-log phase in media containing 0.5 g/L proline as the sole nitrogen source and then after addition of 0.5 g/L glutamine to the medium. (**G**) Co-immunoprecipitations showing an interaction between GFP-Pib2 and Ser33-HA (top panel), but not Gtr1-myc and Ser33-HA (bottom panel). Note we were not able to capture Pib2 from cells exposed to 2 h of amino acid starvation. (**H**) Growth of wild-type, Ser33[S7A], and Ser33[S7D] strains in synthetic medium missing serine and glycine. Cells were grown overnight in SD medium and then diluted into fresh medium missing serine and glycine at the start of the time course. The lines and color-matched shadows show the average and standard deviation from four replicates. Note that all strains are missing Ser3 to isolate the effect of Ser33.

The online version of this article includes the following source data and figure supplement(s) for figure 5:

**Source data 1.** Original western blots for *Figure 5B–G* indicating the relevant bands and treatments.

**Source data 2.** Original files for western blots displayed in *Figure 5*.

**Figure supplement 1.** Rapid Ser33 phosphorylation by TORC1 in response to glutamine stimulation and evidence for a direct interaction with Pib2.

**Figure supplement 1—source data 1.** Original western blots for *Figure 5—figure supplement 1A–C*, indicating the relevant bands and treatments.

**Figure supplement 1—source data 2.** Original files for western blots displayed in *Figure 5—figure supplement 1*.

mutant strains (*Figure 3* and left column, *Figure 4*), while others (targeted by Npr1, a kinase that is repressed by TORC1) are hyperphosphorylated in rapamycin and the mutant strains (left column, *Figure 4—figure supplement 1*). As a result, Par32 exists in a partially phosphorylated state in the SC medium and then shifts to a hyperphosphorylated (slow migrating) state during amino acid starvation (*Hughes Hallett et al., 2014*; *Boeckstaens et al., 2015*) and in *gtr1Δ* and *pib2Δ* cells (*Figure 6B*).

Once we established the Par32 and Sch9 reporter assays, we looked to see if transferring a prototrophic strain from medium that contains an excess of all 20 amino acids (SC), to medium containing a single high-quality nitrogen source (glutamine) or a single poor-quality nitrogen source (leucine or proline), pushes the cell into the predicted Sch9 on and Par32 on, intermediate signaling state (State III). This was not the case: in glutamine medium, the intermediate signaling state was populated for a short time (2.5 and 5 min; *Figure 6C*), but Par32 was dephosphorylated again after 10 min (*Figure 6C*). In leucine medium, we also saw transient population of the intermediate state, but here Sch9 was entirely dephosphorylated after 30 min (*Figure 6C*). Finally, in proline medium, we saw a transition into the complete starvation state (State II), where Par32 is phosphorylated and Sch9 is dephosphorylated (*Figure 6C*).

Next, we looked to see if the intermediate state is populated as cells transition from growth in a poor-quality nitrogen source (proline) to a high-quality nitrogen source (glutamine). This was the case, and to our surprise, TORC1 was already in the intermediate (Sch9 on, Par32 on) signaling state during steady-state growth in proline medium (*Figure 6D*). Thus, it appeared that the TORC1 pathway is driven into a complete starvation state (State II) when cells are first exposed to a poor nitrogen source (2.5–60 min, *Figure 6C*) but then transitions into the intermediate signaling state (State III) as the cells adapt to the poor growth conditions (time 0, *Figure 6D*).

To test this model further, we grew a prototrophic strain in glutamine medium and transferred it to proline medium, but this time followed Par32, Sch9, (and Npr1) phosphorylation for 4 h. As predicted, Sch9 was dephosphorylated initially, but then reactivated (phosphorylated) over time (*Figure 6E*). In contrast, Par32 and Npr1 remained in an active or partially active form (highly phosphorylated and dephosphorylated, respectively) during the entire time course (*Figure 6E*).

We also carried out the same experiment in an *npr1Δ* strain to test the prediction that the adaptation to a poor-quality nitrogen source is driven by the activation of Npr1 and Par32. This was the case, as Sch9 remained dephosphorylated during the entire time course in *npr1Δ* cells (*Figure 6E*).

The observation that cells growing at steady state in SC medium, or glutamine medium, activate Sch9 and inhibit the Npr1-Par32 amino acid starvation response (*Figure 6C and E*), while cells growing in proline medium activate both Sch9 and the Npr1-Par32-dependent amino acid starvation response (*Figure 6E*), also led to another prediction. Wild-type and *npr1Δ* cells should grow at the same rate in SC and glutamine medium since the adaptive amino acid starvation response is off. However, in proline medium, where Npr1-Par32 signaling is active and presumably ensures that

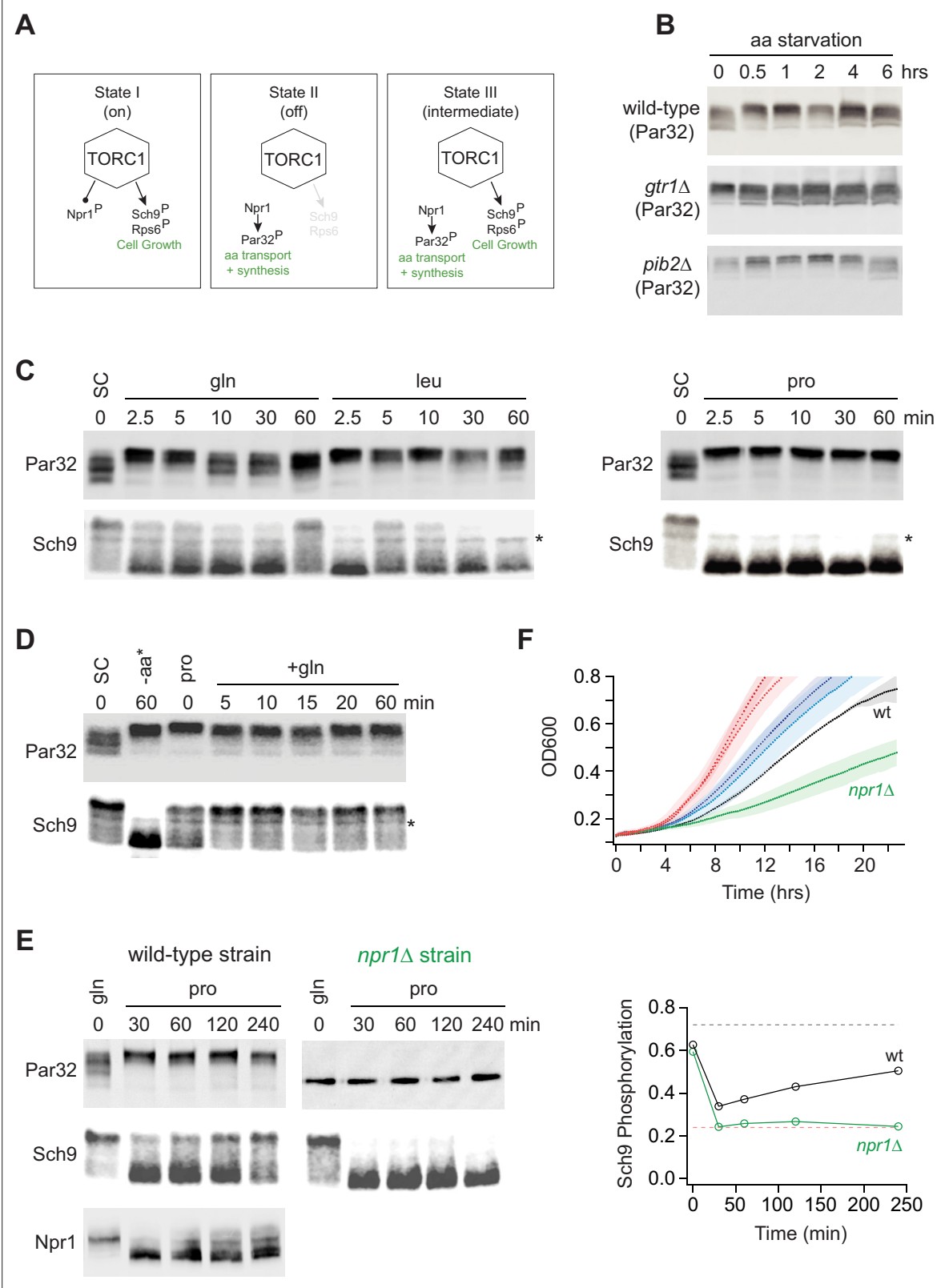

**Figure 6.** Growth in a poor nitrogen source pushes TORC1 into an intermediate signaling state. (**A**) Three-state model of TORC1 signaling as described in the text. (**B**) Par32 phosphorylation measured by SDS-PAGE mobility shift during complete amino acid starvation in wild-type, *gtr1Δ*, and *pib2Δ* strains. (**C**) Par32 and Sch9 phosphorylation in cells grown to mid-log phase in synthetic complete media and then transferred to media containing either glutamine, leucine, or proline as the sole nitrogen source. The asterisk (*) highlights a non-specific band in the western. (**D**) Par32 and Sch9

*Figure 6 continued on next page*

*Figure 6 continued*

phosphorylation in cells grown to mid-log phase in synthetic complete (SC) medium and then exposed to complete amino acid starvation for 60 min (-aa) or grown in medium containing 0.5 g/L proline as the sole nitrogen source (pro) before 0.5 g/L glutamine was added to the culture. (**E**) Par32 and Sch9 phosphorylation in wild-type cells (left panel) and *npr1Δ* cells (right panel) grown to mid-log phase in medium containing 0.5 g/L glutamine as the sole nitrogen source (gln), and then transferred to media containing 0.5 g/L proline as the sole nitrogen source. The graphs on the right show the fraction of Sch9 phosphorylated at each time point, quantified by measuring the fraction of the Sch9 signal that runs above the fastest migrating band. The broken black and red lines show the fraction of Sch9 phosphorylated in SD medium, and after 1 h of rapamycin treatment, respectively. (**F**) Growth of wild-type and *npr1Δ* cells in SC medium (wild-type dark red, *npr1Δ* light red), glutamine medium (wild-type dark blue, *npr1Δ* light blue), and proline medium (wild-type black, *npr1Δ* green). The lines and color-matched shadows show the average and standard deviation from three replicates.

The online version of this article includes the following source data for figure 6:

**Source data 1.** Original western blots for *Figure 6B–E*, indicating the relevant bands and treatments.

**Source data 2.** Original files for western blots displayed in *Figure 6*.

the cells can import/synthesize adequate amounts of nitrogen and each amino acid, the *npr1Δ* cells should grow much slower than wild-type cells. This was also true (*Figure 6F*).

## Tod6 is activated in the intermediate signaling state

The growth curves we collected in SC, glutamine, and proline medium highlight an important fact: yeast grow about four times faster in SC medium than in proline medium (compare red and black lines, *Figure 6F*). However, we see relatively little difference between the phosphorylation level of Sch9—the presumed master regulator of cell growth downstream of TORC1—in those two conditions (*Figure 6D*). This observation led us to posit that one or more of the cell growth control factors is also part of the intermediate (State III) response and acts to slow cell growth in proline medium. The relevant cell growth regulators in yeast are (i) Sfp1, a TORC1-dependent activator of ribosome and protein synthesis gene expression (and thus growth) (*Marion et al., 2004*; *Lempiäinen et al., 2009*); (ii) Tod6, a TORC1- and Sch9-dependent repressor of ribosome and protein synthesis genes (*Lippman and Broach, 2009*; *Huber et al., 2011*); (iii) Stb3, another TORC1- and Sch9-dependent repressor of ribosome and protein synthesis genes (*Huber et al., 2011*; *Liko et al., 2007*); and (iv) Maf1, a TORC1- and Sch9-dependent repressor of RNA Polymerase III and tRNA degradation (*Soulard et al., 2010*). Therefore, to test our model, we searched our phosphoproteomics data to see if any of the aforementioned factors are sensitive to the deletion of Gtr1/2 or Pib2 (and thus likely activated/repressed in the intermediate state). We did not detect Sfp1 in our experiments, but found that Stb3 and Maf1 remain phosphorylated, while Tod6 is dephosphorylated, in the Gtr1/2 and Pib2 delete cells (*Figures 3 and 4*).

Previous studies have shown that TORC1 inhibition triggers the movement of Stb3 and Tod6 into the nucleus where they act to recruit the Rpd3L deacetylase to, and thus repress, the ribosome protein and ribosome biogenesis genes (*Huber et al., 2011*). Therefore, to test if Tod6 is part of the intermediate (State III) response, we measured the localization of Tod6-GFP and Stb3-GFP in a prototrophic strain carrying the nuclear marker Htb2-RFP and growing in SC medium, SC medium + rapamycin, or proline medium. As predicted, Tod6 moved into the nucleus in both proline medium and rapamycin, while Stb3 only moved into the nucleus in rapamycin (*Figure 7*).

Thus, Tod6 is dephosphorylated/activated in proline medium (State III), presumably to help slow the growth rate of the cell in the poor-quality nitrogen source.

## Gtr1/2 and Pib2 signaling during growth in a poor-quality nitrogen source

As a last step in our study, we wanted to see if, and how, changes in Gtr1/2 and Pib2 signaling drive TORC1 into the intermediate state. To address this question, we measured Par32 phosphorylation during the transition from growth in glutamine medium, to growth in proline medium, as we did earlier (*Figure 6E*), but this time in a strain carrying mutations that lock Gtr1 and 2 into their active Gtr1-GTP and Gtr2-GDP-bound forms (Gtr1$^{Q65L}$ and Gtr2$^{S23L}$, Gtr1/2$^{on}$ for short; *Binda et al., 2009*). In the Gtr1/2$^{on}$ strain, Par32 was hyper-phosphorylated during the initial phase of the starvation response (when TORC1 is completely inactive; *Figure 6E*), but then (erroneously) dephosphorylated over time (compare *Figure 6E* and *Figure 8C*), demonstrating that: (i) Gtr1/2 are normally inhibited during steady-state growth in proline, and (ii) Gtr1/2 must remain inactive or partially inactive to keep TORC1

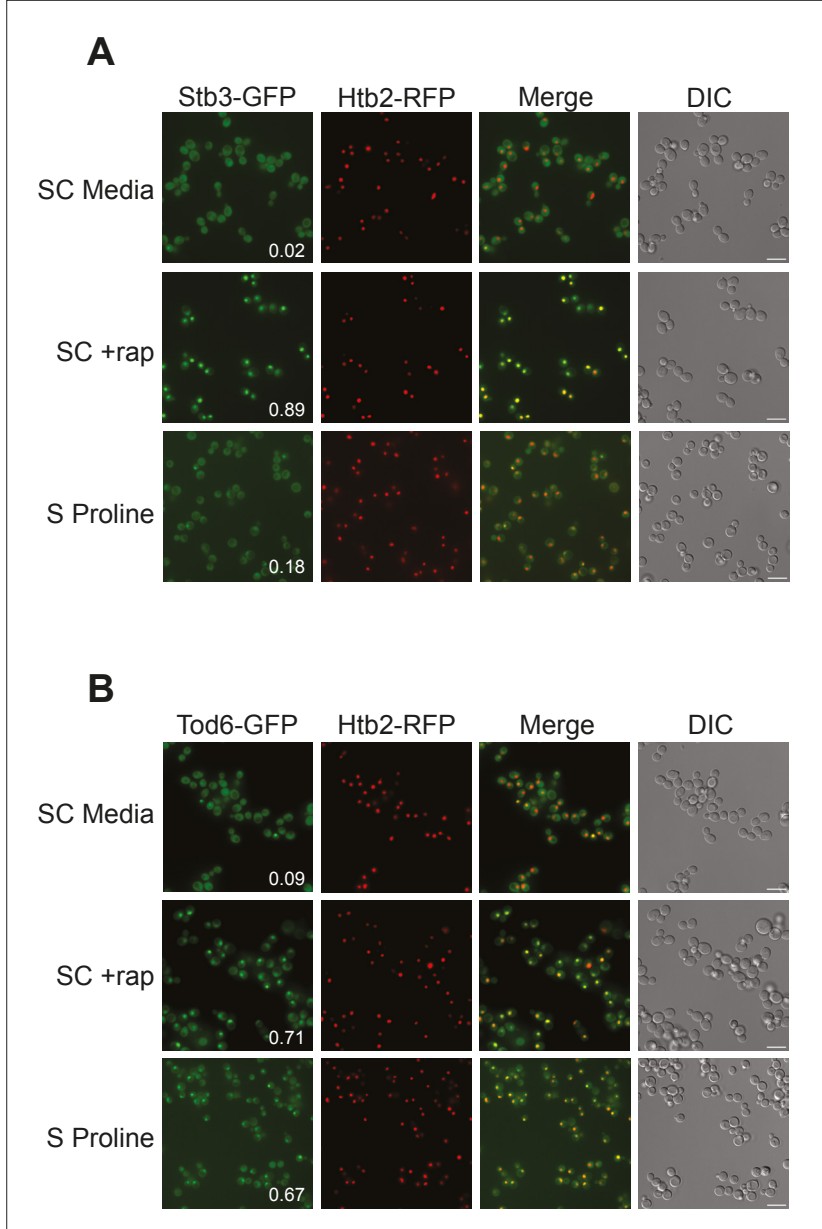

**Figure 7.** Tod6 moves to the nucleus during growth in a poor-quality nitrogen source. Localization of Stb3-GFP (**A**) and Tod6-GFP (**B**) during mid-log phase in the synthetic complete (SC) medium, following exposure to 200 nM rapamycin for 1 h, and during log phase in media containing 0.5 g/L proline as the sole nitrogen source (as labeled). The Tod6-GFP and Stb3-GFP strains are both prototrophic and carry the nuclear marker, Htb2-RFP, at its native locus. Scale bars (in DIC) are 10 μm. The numbers in the Stb3-GFP and Tod6-GFP panels show the fraction of cells with a strong nuclear GFP signal. In the SD and rapamycin control experiments, these values are from a single experiment examining >200 cells. In the proline experiments the values are an average from three biological replicates, with >200 cells per replicate. In those three replicates, Stb3 was nuclear in 21–17%, and 15% (18 ± 3%) of the cells, while Tod6 was nuclear in 70%, 70%, and 60% (67 ± 6%) of the cells.

pathway in the intermediate (Par32 on) signaling state. We also measured the impact that locking Gtr1 and 2 in their inactive Gtr1-GDP and Gtr2-GTP-bound forms (Gtr1/2$^{off}$) has on Rps6 and Sch9 phosphorylation during steady-state growth in proline medium and the transition to growth in gluta-mine medium. Locking Gtr1/2 off caused a moderate decrease in Sch9 (but not Rps6) phosphorylation (*Figure 8A and B*), indicating that Gtr1/2 is partially (rather than fully) inactive during growth in proline medium.

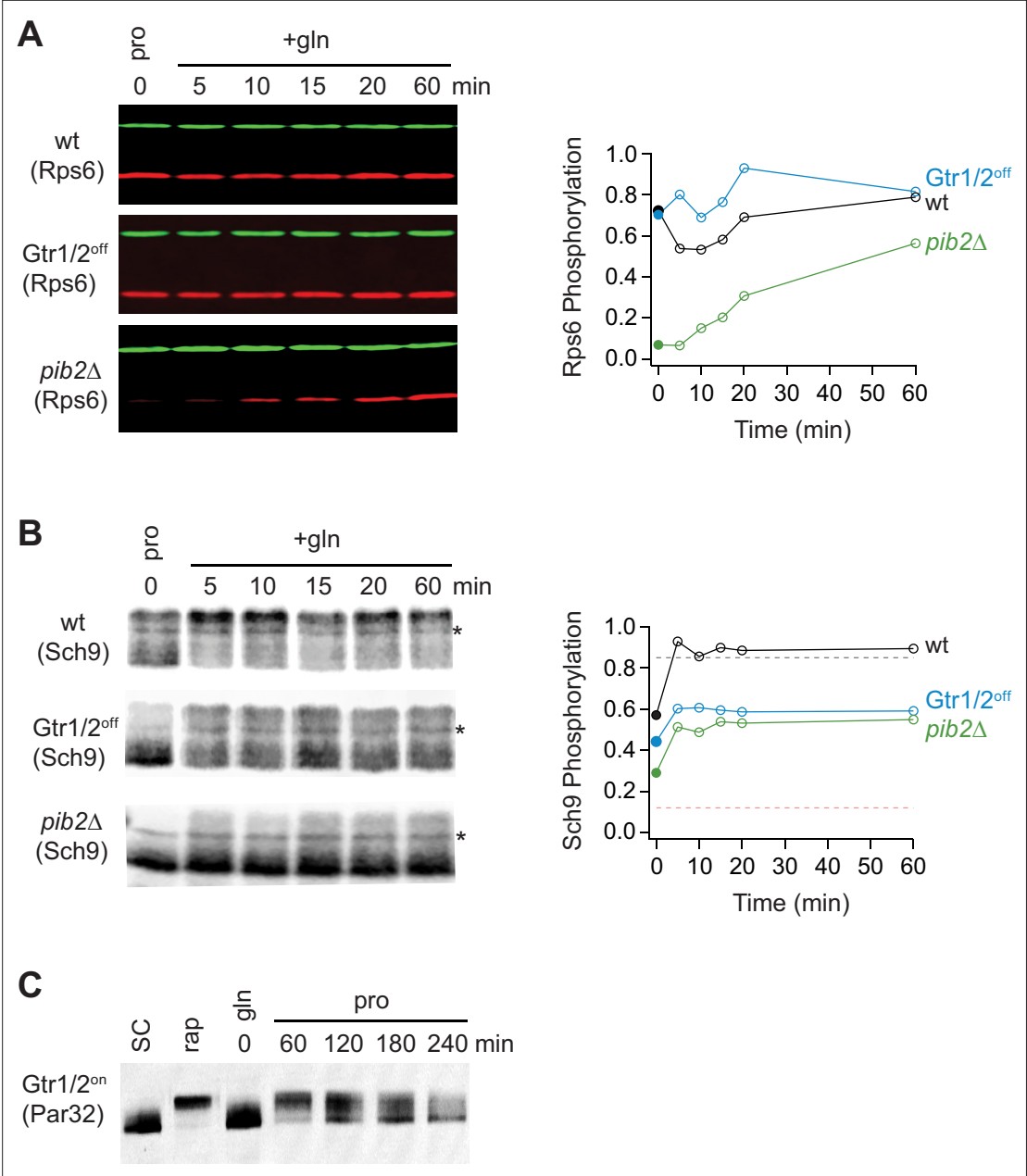

**Figure 8.** Partial Gtr1/2 inactivation drives TORC1 into the intermediate signaling state. (**A**) Rps6 and (**B**) Sch9 phosphorylation, measured in wild-type, GTR1/2off, and *pib2Δ* strains, as cells transition from growth in proline medium to growth in glutamine medium. The data are quantified as described in *Figures 1 and 6E*. Note, wild-type SC samples were included on each gel but were trimmed off the images presented for clarity. (**C**) Par32 phosphorylation as measured by SDS-PAGE mobility shift in GTR1/2on cells, either grown to mid-log phase in synthetic complete (SC) medium then treated with 200 nM rapamycin, or grown to mid-log phase in glutamine medium and then switched to proline medium (compare to *Figure 6E*).

The online version of this article includes the following source data for figure 8:

**Source data 1.** Original western blots for *Figure 8A–C*, indicating the relevant bands and treatments.

**Source data 2.** Original files for western blots displayed in *Figure 8*.

In a parallel set of experiments, we also examined the role that Pib2 plays in regulating TORC1 during growth in proline medium. This was more challenging since relatively little is known about the mechanisms underlying Pib2 signaling. Nevertheless, to assess the role of Pib2 in Npr1-Par32 activation we measured Par32 phosphorylation during the transition from growth in glutamine medium, to growth in proline medium, in a *pib2ΔNID* strain (to lock Pib2 in an active form). Unexpectedly, Par32

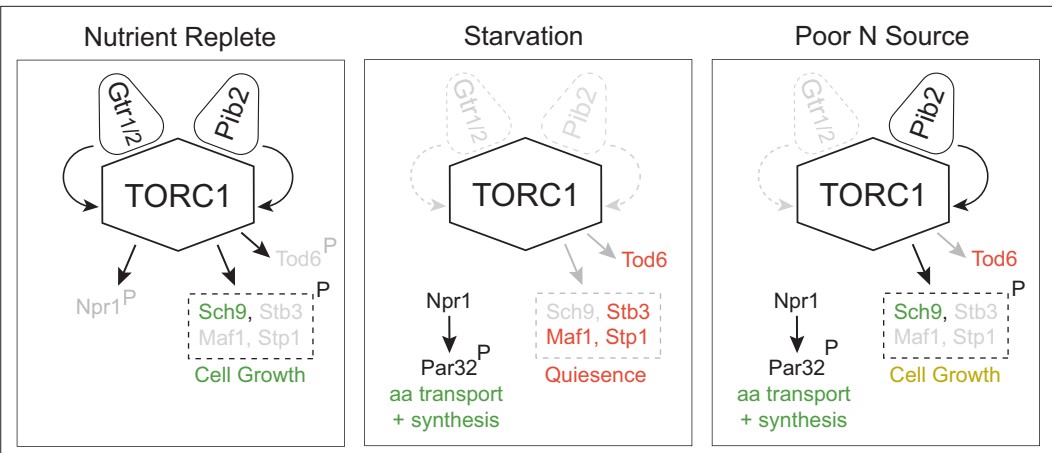

**Figure 9.** Three-state signaling through the TORC1 pathway. Schema describing the three signaling states of the TORC1 pathway, as described in the text. Gray arrows and names show inactive signaling events and proteins. Red names show active repressors; green names show active activators. Sch9, Stb3, Maf1, and Stp1 are phosphorylated in the same conditions and thus shown as a single functional unit (box with broken lines). Rapid cell growth is labeled green, while slow cell growth is labeled yellow.

was completely degraded in the absence of a NID domain in these conditions (data not shown). We have not seen the loss of Par32 in any natural condition, indicating that the NID domain of Pib2 is required for the normal function of the Npr1-Par32 branch of the TORC1 pathway, and that we cannot trap Pib2 in an active state. We then measured the impact that deleting Pib2 has on Rps6 and Sch9 phosphorylation during steady-state growth in proline medium and the transition to growth in glutamine medium. These experiments showed that deleting Pib2 causes a large decrease in Rps6 and Sch9 phosphorylation at time zero (*Figure 8A and B*) and thus that Pib2 is active, or mostly active, during growth in proline medium.

Thus, we conclude that Gtr1/2 is partially off, and Pib2 is on (or mostly on), during steady-state growth in proline medium, and this pushes TORC1 into the intermediate/adaptive (Par32 on, Sch9 on) signaling state.

## Discussion

In this report, we show that the TORC1 pathway takes up at least three distinct signaling states (*Figure 9*). In nutrient-rich medium, TORC1 is fully active and phosphorylates (i) Sch9 and other proteins to drive cell growth, and (ii) Npr1 to suppress the amino acid starvation response (left panel, *Figure 9*). In contrast, when cells are first transferred to medium containing a poor-quality nitrogen source like proline, TORC1 is inhibited (middle panel, *Figure 9*). This blocks cell growth and triggers the activation of Npr1. Then, as cells adapt to the poor-quality nitrogen source (via Npr1), TORC1 is driven into an intermediate activity state where it phosphorylates Sch9 to reinitiate growth, but not Npr1, so that the cell continues to stimulate the increased amino acid transport and synthesis needed to support mass accumulation (right panel, *Figure 9*). Tod6 is also dephosphorylated in the intermediate state, presumably to slow the growth rate of the cell so that it matches the maximum rate achievable in the poor-quality nitrogen source (right panel, *Figure 9*).

The TORC1 pathway is driven into one of the three signaling states described above based on the combination of signals transmitted through Gtr1/2 and Pib2 (*Figure 9*). In nutrient-rich medium, Gtr1/2 and Pib2 are both on and push TORC1 into the fully active state. Then, when cells are first transferred to a poor-quality nitrogen source, Gtr1/2 and Pib2 turn off, or mostly off, triggering strong TORC1 inhibition. Finally, as the cells adapt to the new conditions, Pib2 is reactivated, so that TORC1 can phosphorylate most of the cell growth proteins but not Npr1, Tod6, and other proteins described in *Figure 4*.

This three-state signaling system provides an elegant solution to a problem facing yeast as they transition between nutrient sources. Specifically, when yeast cells grow in nutrient-rich medium they

focus on rapid growth via the import of key amino acids using highly selective transporters (*Bianchi et al., 2019*). As a result, when amino acid levels fall, the cells do not know if there is a low-quality nitrogen source in the extracellular milieu and do not have the capacity to import it even if it is there. Thus, the cells have to inactivate TORC1 and slow/halt their growth. However, once Npr1 is activated, the cells build the capacity to import, and grow using, a wide range of nitrogen-containing compounds (*Schmidt et al., 1998*; *Bianchi et al., 2019*; *De Craene et al., 2001*). Then, if an adequate nitrogen source is available, Pib2 is turned back on so that the cell restarts growth while keeping the Npr1-dependent starvation response (and the import of alternate nitrogen sources) active.

Important implications of this three-state model include that (i) yeast cells (likely including fungal pathogens) can be pushed to import alternative nitrogen sources, including toxic amino acid analogs and related compounds, by deletion or inhibition of Gtr1/2 or Pib2, and (ii) yeast cell growth on poor nitrogen sources can be halted by deletion of, or inhibition of, Pib2.

The data presented here also reveal another unique aspect of the Gtr1/2 and Pib2 control circuit, namely that the activity of Ser33 (and its homolog of Ser3) depends on signaling through TORC1-Pib2, and that this in turn appears to ensure that Ser33 is only phosphorylated in the presence of a high-quality nitrogen source. Our data suggests that this occurs due to direct binding of Ser33 to Pib2, leading to a model that is reminiscent of the RagC-dependent binding and regulation of TFEB in human cells, where TFEB is regulated in response to amino acid signals (via RagC), but not the hormone signals transmitted to TORC1 through Rheb (*Cui et al., 2023*; *Napolitano et al., 2020*). We hypothesize that the TORC1-Pib2 connection to Ser33 is setup this way to couple Ser33 activity to the level of glutamate/glutamine in the cell (rather than overall amino acid levels and other stress signals that regulate Gtr1/2) since the second step in serine synthesis involves the transfer of an amine group from glutamate to the Ser33 product, 3-phosphohydroxypyruvate (*Mattaini et al., 2016*).

Beyond the implications this work has for understanding TORC1 signaling in yeast, our study also provides important insight into the design (and analysis) of other signaling pathways, including mTOR. Previous experiments examining TORC1 signaling in *S. cerevisiae*, including our own, focused on measuring TORC1 activity using one or two downstream reporters (usually pRps6 or pSch9). This led to the view that Gtr1/2 and Pib2 act as redundant activators of TORC1 (*Hatakeyama, 2021*). In reality, however, Gtr1/2 and Pib2 are semi-redundant activators of TORC1, and drive TORC1 into a fully active, partially active, or inactive state via on/on, off/on, or off/off modes of Gtr1/2 and Pib2 signaling. Fully active TORC1 then phosphorylates all of its downstream targets, while partially active TORC1 phosphorylates a subset of its targets, to create a condition-dependent response. We argue that other stress and starvation signaling pathways likely work in a similar way; it is just that multilevel signaling has been overlooked since it is difficult to detect using standard reporter assays.

## Materials and methods
### Strain construction
All of the strains in this study were made in haploid (W303) *S. cerevisiae*, using standard methods (*Storici and Resnick, 2006*; *Storici et al., 2001*), are listed in *Supplementary file 3* (note there are two tabs), and are available upon request. For strains grown in single nitrogen source media, we restored prototrophy using single copy plasmids containing the required complements (*LEU2, HIS3,* and/or *URA3*) and by integrating the *TRP1* gene into the genome (*Mülleder et al., 2016*).

### Synthetic medium
The experiments in *Figures 1–4* and most of *Figure 5* used standard auxotrophic (His-, Leu-) lab strains. These cells were grown in SC medium containing ammonium sulfate and all 20 amino acids, and then switched to the same medium missing one or more amino acid, as indicated. The experiments in *Figures 5F and 6–8* used prototrophic yeast strains, and cells were grown and studied in synthetic medium without ammonium sulfate, supplemented with a single nitrogen source (e.g., Leu, Gln, or Pro) unless noted (SC panels, *Figure 7*).

### Rps6 phosphorylation assay
Cultures were grown in conical flasks, shaking at 200 rpm and 30°C until mid-log phase (OD$_{600}$ 0.4–0.7). At that point, a 47 mL sample was collected, mixed with 3 mL of 100% trichloroacetic acid (TCA), and

held on ice for at least 30 min (but no more than 6 h). The remaining culture was then collected by filtration and transferred to SC-amino acids, SC-leu, or SC-glutamine medium after two washes with 100 mL of the same medium, and additional samples collected as described above. Cells exposed to glutamine starvation were also treated with a freshly made 2 mM methionine sulfoximine (MSX, Sigma-Aldrich 15985-39-4). The TCA- precipitated samples were then centrifuged at 4000 rpm for 5 min at 4°C, washed twice with 4°C water, twice with acetone, and disrupted by sonication (2×) at 15% amplitude for 5 s before centrifugation at 12,000 rpm for 30 s. The cell pellets were then dried in a speedvac for 10 min at room temperature and frozen until required at −80°C.

Protein extraction was performed by bead beating (6×1 min, full speed) in urea buffer (6 M urea, 50 mM Tris–HCl pH 7.5, 5 mM ethylenediaminetetraacetic acid [EDTA], 1 mM phenylmethylsulfonyl fluoride, 5 mM NaF, 5 mM NaN$_3$, 5 mM NaH$_2$PO$_4$, 5 mM $p$-nitrophenylphosphate, 5 mM β-glycero-phosphate, and 1% SDS) supplemented with complete protease and phosphatase inhibitor tablets (Roche, Indianapolis, IN; 04693159001 and 04906845001). The lysate was then harvested by centrifugation for 5 min at 3000 rpm, resuspended into a homogeneous slurry, and heated at 65°C for 10 min. The soluble proteins were then separated from insoluble cell debris by centrifugation at 12,000 rpm for 10 min, and the lysate stored at −80°C until required.

For protein phosphorylation analysis, the protein extracts were run on a 12% acrylamide gel and transferred to a nitrocellulose membrane. Western blotting was then carried out using anti-pRPS6 antibody (Cell Signaling, 4858) at a 1/2500 dilution, and anti-PGK1 antibody (Invitrogen, 459250) at a 1/10,000 dilution, and anti-mouse and anti-rabbit secondaries, labeled with an IRDye 680RD (LI-COR 926-68071) and IRDye 800CW (LI-COR 926-32210) both at a 1/10,000 dilution, and the blots scanned using a LI-COR Odyssey Scanner (LI-COR, Lincoln, NE). Band intensities were quantified using the LI-COR Image Studio Software.

## SYTOX cell death assay

Cells were inoculated into a 10 mL starter culture, and then reinoculated into SC medium to ensure at least 12 h of logarithmic growth. Once the cells had reached an OD$_{600}$ between 0.4 and 0.7, they were collected by filtration, washed with 25 mL of SC medium missing glutamine, and supplemented with 2 mM methionine sulfoximine (SC-gln), and then resuspended in 25 mL of the same medium for 6 h. The cells were then moved into 8-well microslides (Ibidi, 80826) pretreated with Concanavalin A (Fisher Scientific ICN15071001) and stained with SYTOX Green Fluorescent Dye (Invitrogen S7020). Specifically, after the cells settled in the well, they were washed with 50 mM sodium citrate and then treated with 50 mM sodium citrate containing 1 mM SYTOX Green Dye for 30 min (in the dark). The cells were then washed two additional times with 50 mM sodium citrate, resuspended in the same buffer, and fluorescence and DIC images acquired using a Nikon Eclipse Ti-E microscope equipped with a ×100 objective and a Photometrics Prime 95B camera (excitation at 488 nm, emission at 515 nm, 1 s exposure).

## Sch9, Par32, and Npr1 mobility shift experiments

Cells examined in the band shift (gel mobility) experiments were grown to OD$_{600}$ 0.4–0.7 in SC medium, or SC medium missing amino acids but containing proline, glutamine, or leucine, as indicated. Samples were then collected by TCA precipitation and lysed in urea buffer as described above for Rps6. Sch9-3xHA samples were also subjected to cleavage by 2-nitro-5-thiocyanatobenzoic acid (NTCB) in N-cyclohexyl-2-aminoethanesulfonic acid (CHES, pH 10.5) overnight at 30°C (1 mM NTCB and 100 mM CHES) as described previously (*Hughes Hallett et al., 2014*; *Urban et al., 2007*).

Par32-13xMyc and Npr1-3XFLAG samples (25 and 100 µg of total protein, respectively) were run on a 7.5% acrylamide gel for 3 h at 80 V and transferred to a nitrocellulose membrane. Western blotting was then carried out using anti-Myc (Thermo Scientific MA12316) or anti-FLAG (Sigma-Aldrich F1804-1MG) antibodies at a 1/1000 dilution, and an anti-mouse secondary labeled with an IRDye 800CW (LI-COR 926-32210), at a 1/10,000 dilution, and the blots scanned using a LI-COR Odyssey Scanner (LI-COR).

Sch9-3XHA samples were (10 µg of total protein, respectively) run on a 12% acrylamide gel for 3 h at 80 V and transferred to a nitrocellulose membrane. Western blotting was then carried out using an anti-HA antibody (Sigma-Aldrich, 11583816001) at a 1/1000 dilution, and an anti-mouse secondary,

labeled with an IRDye 800CW (LI-COR 926-32210), at a 1/10,000 dilution, and the blots scanned using a LI-COR Odyssey Scanner (LI-COR).

## Ser33 PhosTag bandshift experiments

Ser33-3xFLAG cells were grown and harvested as described above for the Rps6 experiments. 25 µg of Ser33-3xFLAG protein extract (per lane) was then loaded on an 8% $Zn^{2+}$/PhosTag BisTris gel at 60 V for 3 h and 45 min in MOPS running buffer made following manufacturer's instructions (FUJIFILM, Wako AAL-107). To prevent the 5 mM EDTA in our urea buffer from disrupting band migration, we added 2 uM $ZnNO_3$ to our loading buffer (EDTA-free) prior to mixing it with our samples. After electrophoresis, the gels were washed with transfer buffer containing 5 mM EDTA for 10 min and then again with standard transfer buffer (no EDTA) for 10 min. Western blotting then proceeded as described above.

## Phosphoproteomics

The cells examined using phosphoproteomics were collected using TCA precipitation, as described above. All remaining steps used MS-grade reagents (including water).

First, the cell pellets were resuspended in 400 µL of MS-urea buffer (8 M urea, 100 mM ammonium bicarbonate [ABC], 5 mM EDTA). Proteins were then extracted by bead beating (as described above) and eluted into wide-mouth tubes (without a 65°C denaturation step) and the final protein concentration in each sample measured using a BCA assay.

200 µg of total protein was taken from each sample and diluted to 1 µg/µL in a 2.0 mL low-bind tube (Thermo Scientific 88379) using the 8 M urea buffer above. The samples were alkylated and reduced by treatment with 5 mM tris(2-carboxyethyl)phosphine hydrochloride and 5 mM iodoacetamide at room temperature for 30 min. The reduced and alkylated samples were then diluted by adding 70 µL of 50 mM ABC (so that the final urea concentration was 5.5 M) and 20 ng/µL of LysC (New England Biolabs, P8109S) to each sample, and digested at 37°C, shaking at 700 rpm, for 3 h. The samples were then diluted again with 1.3 mL 50 mM ABC to bring the urea concentration to 1 M, and treated with 2 µg trypsin (Promega, v511c), shaking overnight at 700 rpm, and 37°C.

The next morning trypsinization was quenched by adding TFA to a final concentration of 1% v/v, and the samples clarified by centrifugation at 15,000 rpm for 5 min. The peptide mix was then desalted using Sep Pak Plus C18 cartridges (Waters: WAT020515) on a vacuum manifold. First, cartridges were equilibrated by flushing with 5 mL of solution B (65% acetonitrile [MeCN], 0.1% trifluoroacetic acid [TFA]) and then 10 mL solution A (2% MeCN, 0.1% TFA). The peptide samples, now around 2 mL in volume, were then diluted with 8 mL of solution A and slowly run through the cartridges. The columns were then washed with 10 mL of solution A and then peptides eluted twice using 600 µL of solution B and collected in a low-bind tube. The peptides were then dried in a speed vac at room temperature and stored at –80°C.

Phosphopeptides were enriched using magnetic Ti(IV)-IMAC beads (MagReSyn MR-TIM005) following the manufacturer's instructions. Specifically, 40 µL beads were equilibrated using three washes with loading buffer (0.1 M glycolic acid in 80% MeCN, 5% TFA). Dried peptide samples were then resuspended in 200 µL loading buffer and incubated with the beads for 20 min, shaking at 600 rpm and room temperature. The beads were then washed with 200 µL of loading solution, 100 µL wash solution 1 (80% ACN, 1% TFA), and 100 µL wash solution 2 (10% ACN, 0.2% TFA) for 2 min each, with 600 rpm agitation. The phosphopeptides were then eluted twice using 135 µL of 1% $NH_4OH$ into 90 µl of 10% formic acid (FA), leading to a final volume of 360 µL.

The purified phosphopeptides were then desalted on micro spin columns (Nest Group: SEM SS18V). Each step used centrifugation at 1500 × $g$ for 1 min. First, columns were conditioned with 400 µL 90% MeCN, 0.1% FA and then equilibrated with 350 µL 5% MeCN, 0.1% FA. The samples were then loaded onto the columns, washed with 350 µL 5% MeCN, 0.1% FA, eluted in 200 µL 50% MeCN, 0.1% FA, dried using a speedvac at room temperature, and stored at –80°C.

## Mass spectrometry

Samples were resuspended in 10.5 µL 0.1% FA, and 1.5 µL of the suspension injected for HPLCESI-MS/MS analysis. Data acquisition was performed in positive ion mode on a Thermo Scientific Orbitrap Fusion Lumos tribrid mass spectrometer fitted with an EASY-SpraySource (Thermo Scientific, San Jose, CA). NanoLC was performed using a Thermo Scientific UltiMate 3000 RSLCnano System with an

EASY Spray C18 LC column (Thermo Scientific, 50 cm × 75 µm inner diameter, packed with PepMap RSLC C18 material, 2 µm, cat. # ES803): loading phase for 15 min at 0.300 µL/min; linear gradient of 1–34% Buffer B in 119 min at 0.220 µL/min, followed by a step to 95% Buffer B over 4 min at 0.220 µL/min, hold 5 min at 0.250 µl/min, and then a step to 1% Buffer B over 5 min at 0.250 µL/min and a final hold for 10 in (total run 159 min); Buffer A=0.1% FA; Buffer B=0.1% FA in 80% ACN. Spectra were collected using XCalibur, version 2.3 (Thermo Fisher Scientific). Precursor scans were acquired in the Orbitrap at 120,000 resolution on a mass range from 375 to 1575 Th. Precursors were isolated with an isolation width of 1.6 Th and subjected to higher energy collisional dissociation. MS/MS scans were acquired in the ion trap on the m/z range of 120–2000 Th with a fill time of 35 ms.

### Phosphoproteomic data analysis

Progenesis QI for proteomics software (version 2.4, Nonlinear Dynamics Ltd., Newcastle upon Tyne, UK) was used to perform ion-intensity based label-free quantification as described previously. In brief, in an automated format, .raw files were imported and converted into two-dimensional maps (y-axis=time, x-axis=m/z) followed by selection of a reference run for alignment purposes. An aggregate data set containing all peak information from all of the samples in a given experiment was created from the aligned runs, which was then further narrowed down by selecting only +2, +3, and +4 charged ions for further analysis. A peak list of fragment ion spectra was exported in Mascot generic file (.mgf) format and searched against a UniProt *S. cerevisiae* S288c database (6728 entries) using Mascot (Matrix Science, London, UK; version 2.6). The search variables that were used were 10 ppm mass tolerance for precursor ion masses and 0.5 Da for product ion masses; digestion with trypsin; a maximum of two missed tryptic cleavages; variable modifications of oxidation of methionine and phosphorylation of serine, threonine, and tyrosine; 13C=1. The resulting Mascot.xml file was then imported into Progenesis, allowing for peptide/protein assignment, while peptides with a Mascot Ion Score of <25 were not considered for further analysis.

Peptide ion data were exported as a .csv file. Positions of phosphorylation sites within the protein were obtained by mapping the peptide sequence to the protein sequence in the .fasta file used for the database search. Duplicate entries of the same peptide ion mapping to more than one protein were collapsed to one entry. Normalized intensities of phosphorylated peptides mapping to the same phosphosite were summed together.

### Crosslinking and coimmunoprecipitation

Strains carrying GFP-Pib2 and 3xHA-Ser33, or Gtr1-13xMyc and 3xHA-Ser33, were grown in 500 mL of synthetic complete media to log phase as described above. 250 mL of each sample was collected by filtration and snap frozen in liquid nitrogen. The remaining samples were then captured by filtration, washed with 250 mL of SC medium missing amino acids (-aa), transferred to 250 mL of SC-aa media for 2 h, and collected by filtration and snap freezing. To begin protein extraction, frozen cell pellets were washed with 5 mL of 4°C Immunoprecipitation Lysis Buffer (IPLB; 20 mM 4-(2-hydroxyethyl)–1-piperazineethanesulfonic acid, pH 7.5, 150 mM potassium acetate, 2 mM magnesium acetate, 1 mM ethylene glycol bis(2-aminoethyl)tetraacetic acid, and 0.6 M sorbitol) (*Murley et al., 2017*) and then resuspended in 1 mL of IPLB supplemented with complete protease and phosphatase inhibitors (IPLB++). Samples resuspended in IPLB++ were split between two 2 mL screw-cap tubes, sheared by bead beating, and eluted into 1.5 mL wide-mouth tubes as described above. Lysates were homogenized by gentle vortexing and combined into a fresh 2 mL tube. Crosslinking was performed by treating the lysates with 0.25 µM of dithiobis(succinimidyl propionate) (DSP) at 4°C for 30 min with gentle rotation. The reaction was then quenched by adding 100 mM Tris, pH 7.5, and the sample held on ice for 30 min. Cell membranes were then solubilized by adding digitonin to a concentration of 1% with gentle rotation for 30 min and clarified by centrifugation at 12,000 rpm at 4°C for 10 min, and the supernatants transferred to a fresh 2.0 mL tube.

To co-purify GFP-Pib2 or Gtr1-13xMyc and any interacting Ser33, 50 µL of µMACS anti-GFP or anti-c-Myc (Miltenyi Biotec, 130-091-125 and 130-091-123) was added to clarified extract and incubated at 4°C while rotating for 2 h. µMACS columns were equilibrated by adding 200 uL of IPLB++ containing 1% digitonin. Samples were then added to each column (on a magnet) and allowed to pass through by gravity. The beads were then washed three times with 200 µL of IPLB++ containing 0.1% digitonin, and then twice with 500 µL IPLB (without digitonin). The protein was then eluted, first by incubation

for 5 min with 20 μL elution buffer (supplied with μMACS kit) pre-heated to 95°C, and then by adding 2 × 40 μL of the same buffer.

## Immunoprecipitation and MS-phosphomapping of Ser33

Immunoprecipitation of Ser33-3xFLAG was performed using the protocol described above, but without the DSP crosslinking, Tris quenching, and digitonin membrane-permeabilization steps. Instead, IPLB buffer was supplemented with 0.25% TWEEN, and 3xFLAG-tagged Ser33 was purified using anti-FLAG conjugated antibodies (130-101-591). Immunopurified samples were then separated by SDS-PAGE gel, and gel slices around bands corresponding to the correct molecular weight of Ser33 were excised and sent for phosphoproteomic analysis by mass spectrometry.

Gel slices were washed for 15 min each with water, 50/50 MeCN/water, MeCN, 100 mM ABC, followed by 50/50 MeCN/100 mM ABC. The solution was then removed and the gel slices dried by vacuum centrifugation. Next, the dried gel slices were reduced by covering them with 10 mM dithiothreitol in 100 mM ABC and heating them at 56°C for 45 min; alkylated by covering them with a solution of 55 mM iodoacetamide in 100 mM ABC and incubating in the dark at ambient temperature for 30 min, and washed with 100 mM ABC for 10 min and 50 mM ammonium biocarbonate +50% MeCN for 10 min. The gel slices were then dried again and treated with an ice-cold solution of 12.5 ng/μL trypsin (Promega, Madison, WI) in 100 mM ABC. After 45 min, the trypsin solution was removed, discarded, and a volume of 50 mM ABC was added to cover the gel slices, and they were incubated overnight at 37°C with mixing on a shaker. Samples were then spun down in a microfuge and the supernatant collected. The same gel slices were then incubated in 0.1% TFA and MeCN, centrifuged, and the supernatant collected. At this point, the digestion supernatant and the extraction supernatant were pooled, split into two tubes, and concentrated using vacuum centrifugation. One tube was further digested with thermolysin (Promega) by resuspending the tryptically digested peptides with a solution containing 50 mM Tris-HCl pH 8 and 0.5 mM calcium chloride and adding 1 μg of thermolysin. Digestion was carried out at 75°C with mixing for 5 h. The thermolysin was quenched by adding TFA to a 0.5% final concentration. All samples were desalted using ZipTip $C_{18}$ (Millipore, Billerica, MA) and eluted with 70% MeCN/0.1% TFA. The desalted material was concentrated to dryness in a speed vac.

The proteolytically digested samples were brought up in 20 μL of 2% MeCN in 0.1% FA and 18 μL and then analyzed by LC/ESI MS/MS with a Thermo Scientific Easy-nLC II (Thermo Scientific, Waltham, MA) coupled to an Orbitrap Elite ETD (Thermo Scientific) mass spectrometer using a trap-column configuration as described in *Licklider et al., 2002*. In-line de-salting was accomplished using a reversed-phase trap column (100 μm×20 mm) packed with Magic $C_{18}$AQ (5 μm, 200 Å resin; Michrom Bioresources, Auburn, CA) followed by peptide separations on a reversed-phase column (75 μm×250 mm) packed with Magic $C_{18}$AQ (5 μm, 100 Å resin; Michrom Bioresources) directly mounted on the electrospray ion source. A 45 min gradient from 2% to 35% MeCN in 0.1% FA at a flow rate of 400 nL/min was used for chromatographic separations. A spray voltage of 2500 V was applied to the electrospray tip, and the Orbitrap Elite instrument was operated in the data-dependent mode, switching automatically between MS survey scans in the Orbitrap (AGC target value 1,000,000, resolution 120,000, and injection time 250 ms) with collision-induced dissociation MS/MS spectra acquisition in the linear ion trap (AGC target value of 10,000 and injection time 100 ms), higher-energy collision-induced dissociation (HCD) MS/MS spectra acquisition in the Orbitrap (AGC target value of 50,000, 15,000 resolution, and injection time 250 ms) and electron transfer dissociation (ETD) MS/MS spectra acquisition in the Orbitrap (AGC target value of 50,000, 15,000 resolution, and injection time 250 ms). The three most intense precursor ions from the Fourier-transform (FT) full scan were consecutively selected for fragmentation in the linear ion trap by CID with a normalized collision energy of 35%, fragmentation in the HCD cell with normalized collision energy of 35%, and fragmentation by ETD with 100 ms activation time. Selected ions were dynamically excluded for 30 s.

Data analysis was performed using Proteome Discoverer 1.4 (Thermo Scientific). The data were searched against the *Saccharomyces* Genome Database (downloaded 2/03/2011; https://www.yeast-genome.org/) that was appended with protein sequences from the common Repository of Adventitious Proteins or cRAP (https://www.thegpm.org/crap/). Two searches were performed corresponding to the proteolytic enzymes trypsin and trypsin/thermolysin (no enzyme selected). The maximum missed cleavages were set to 2. The precursor ion tolerance was set to 10 ppm, and the fragment ion tolerance was set to 0.8 Da. Variable modifications included oxidation on methionine (+15.995 Da),

carbamidomethyl (+57.021 Da) on cysteine, and phosphorylation on serine, threonine, and tyrosine (+79.996 Da). Sequest HT was used for database searching. PhosphoRS 3.1 (*Taus et al., 2011*) was used for assigning phosphosite localization probabilities. All search results were run through PSM Validator for false discovery rate evaluation of the identified peptides.

## Fluorescence microscopy

For microscopy in SC + proline, cells were grown in 25 mL over 3 days, refreshing the media by removing half and readding equivalent volume each day until they reached the logarithmic growth phase. Cells grown in SC were inoculated into a 10 mL starter culture, and then diluted again into a 25 mL overnight culture. In both cases, cells were grown to log phase ($OD_{600}$ 0.4–0.6) and then pipetted onto 8-well microslides (Ibidi, 80826) that had been pretreated with Concanavalin A. Cells were then washed with matching media (SD + proline or SC) and then imaged using a Nikon Eclipse Ti-E microscope equipped with a ×100 objective and a Photometrics Prime 95B camera. GFP images were acquired with an excitation of 488 nm and an emission of 515 nm using 1 s exposure. RFP images were acquired with an excitation of 561 nm and an emission of 632 nm using a 500 ms exposure. DIC images were captured using a 60 ms exposure. To examine the impact of complete TORC1 inhibition, cells growing in SC medium were treated with 200 mM rapamycin for 60 min (within the well) and then imaged again with the same settings. Images were analyzed and quantified in ImageJ (*Schindelin et al., 2012*).

## Proliferation assays

Three biological replicates of each strain were grown overnight and aliquoted to tubes containing 1 $OD_{600}$ unit. Samples were then washed with 1 mL filtered water, and then resuspended in 1 mL filtered water. 100 μL of culture were diluted into 900 μL of each medium, bringing the starting concentration to an $OD_{600}$ of 0.1. For each biological replicate, three technical replicates were plated across a 96-well plate (Corning 3370). Plates were covered with Breathe-Easy membranes (Z380059 Sigma-Aldrich), and growth was measured on a TECAN Infinite M Nano plate reader. Growth settings were orbital shaking with a 5 mm amplitude for 8 min followed by a linear shake with a 5 mm amplitude for 2 min at 30°C. $OD_{600}$ readings were taken every 10 min for 23 h.

## Acknowledgements

We thank Claudio De Virgilio for sharing GTR1 and 2 mutant plasmids, and Kyle Cunningham for sharing the GFP-Pib2 plasmid, used to make our mutant strains. We also thank Phil Gafken and Lisa Jones of the Fred Hutchinson Cancer Research Center's Proteomics Facility for carrying out the Ser33 peptide mapping experiments. This work was supported by the National Institutes of Health (NIH) grants R01GM097329 and T32GM136536.

## Additional information

### Funding

| Funder | Grant reference number | Author |
| --- | --- | --- |
| National Institute of General Medical Sciences | R01GM097329 | Andrew P Capaldi |
| National Institute of General Medical Sciences | T32GM136536 | Jacob H Cecil Cristina M Padilla |

The funders had no role in study design, data collection and interpretation, or the decision to submit the work for publication.

### Author contributions

Jacob H Cecil, Conceptualization, Formal analysis, Investigation, Writing – original draft, Writing – review and editing; Cristina M Padilla, Formal analysis, Investigation, Writing – review and editing; Austin A Lipinski, Formal analysis, Methodology, Writing – review and editing; Paul Langlais, Formal

analysis, Supervision, Writing – review and editing; Xiangxia Luo, Formal analysis, Writing – review and editing; Andrew P Capaldi, Conceptualization, Formal analysis, Supervision, Funding acquisition, Investigation, Methodology, Writing – original draft, Project administration, Writing – review and editing

**Author ORCIDs**
Jacob H Cecil ⓘ https://orcid.org/0000-0002-1227-8777
Cristina M Padilla ⓘ https://orcid.org/0009-0000-3330-9782
Andrew P Capaldi ⓘ https://orcid.org/0000-0002-7902-2477

Reviewer #1 (Public Review): https://doi.org/10.7554/eLife.94628.2.sa1
Reviewer #2 (Public Review): https://doi.org/10.7554/eLife.94628.2.sa2
Reviewer #3 (Public Review): https://doi.org/10.7554/eLife.94628.2.sa3
Author response https://doi.org/10.7554/eLife.94628.2.sa4

## Additional files

### Supplementary files
MDAR checklist

Supplementary file 1. Phosphoproteomic data for wild-type, Gtr1/2 delete and Pib2 delete strains.

Supplementary file 2. Summary of rapamycin dependent phosphorylation sites in Ser33.

Supplementary file 3. Strain table.

Supplementary file 4. Raw data mapping Ser33 phophorylation sites (Part I).

Supplementary file 5. Raw data mapping Ser33 phophorylation sites (Part II).

Supplementary file 6. Raw data mapping Ser33 phophorylation sites (Part III).

Supplementary file 7. Raw data mapping Ser33 phophorylation sites (Part IV).

### Data availability
All data generated and analyzed during this study are included in the manuscript and the supporting files.

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
