## [Editor Report · eLife assessment]

The study presents **valuable** findings concerning how a highly conserved signal transduction pathway helps budding yeast cells adapt their growth to nitrogen sources of differing qualities. However, the evidence is **incomplete** for the authors' main claim that the pathway adopts three distinct states depending on the nitrogen source. The presented data, particularly phospho-proteomic datasets, will be of interest to the cell growth signaling community.

---

## [Referee Report · Reviewer #1 (Public Review)]

Summary:

TOR complex 1 (TORC1) is a key regulator cell growth in response to nutrients, and it therefore integrates inputs from multiple nutrient-sensing regulators. However, we still do not understand how each upstream regulatory branch contributes to TORC1 activity under different nutrient conditions. The authors set out to answer this question using budding yeast (*Saccharomyces cerevisiae*) as a model eukaryote. Yeast TORC1 is activated by two upstream regulators: the highly conserved GTPases Gtr1/2 and the PI3P-binding protein Pib2. The cooperation of these regulators towards TORC1 activation has been unclear, with some studies suggesting that they act in parallel (i.e. redundantly), and others suggesting a more complex picture. By exploring the dependence of different TORC1 substrates on Gtr1/2 and Pib2 activity, the authors have discovered that Gtr1/2 and Pib2 do not act redundantly, but instead are part of a mechanism that drives the TORC1 pathways into three distinct activity levels: (i) both Gtr1/2 and Pib2 ON in rich nutrients (leading to the highest TORC1 activity), (ii) Gtr1/2 OFF and Pib2 ON in poor quality nitrogen sources (intermediate TORC1 activity), and (iii) both Gtr1/2 and Pib2 OFF under starvation conditions (lowest TORC1 activity).

Strengths:

The relation between Gtr1/2 and Pib2 has remained a mystery for a long time, making it difficult to interpret the results of experiments in which one of the two regulators is inactive or missing. By employing a phosphoproteomics assay, the authors were able to monitor the phosphorylation of multiple TORC1 substrates in response to TORC1 inhibition (via rapamycin) and in mutants carrying deletions of Gtr1/2 or Pib2. In this way, they could identify two groups of substrates: those that require the activity of both regulators, and those that remain active when a single regulator is active. These data clearly demonstrate the non-redundancy of the Gtr1/2 and Pib2, especially since the different groups of substrates seem to correspond to groups of proteins with distinct functions.

Weaknesses:

- The first section of the Results contains an analysis of Gtr1/2- and Pib2-dependent signaling using Rps6 as a TORC1 reporter. I do not think that Rps6 is an appropriate readout for this type of work, as it is not a direct TORC1 substrate, and it also lies downstream of TORC2 [Yerlikaya et al. 2016]. The authors obtain several puzzling results with Rps6, and later on (pg. 8) remark that the level of Rps6 phosphorylation does not always correspond to TORC1 activity. While this is an interesting finding in its own right and will certainly be interesting for the yeast TOR community, I do not see why the Results need to open with such a confusing section, and why Rps6 features so prominently throughout the manuscript.

- There is very large ambiguity regarding the types of media and strains that are used (prototrophic vs auxotrophic). The authors use SC medium which, if I understand correctly, contains ammonium and a supplement of amino acids. They then use single amino acid dropouts (e.g. SC -gln and SC -leu) to probe TORC1 activity under "partial starvation" conditions. However, the cells are anything but starved in these experiments, and I do not know how to interpret results obtained with such media. Even when amino acids are completely removed, the cells are still able to grow on ammonium. The matter gets further complicated because it appears that the authors use prototrophic strains with single nitrogen source media, but not with complete or "partial starvation" media. Since this study aims to elucidate the roles of nutrient-sensing regulators upstream of TORC1, I would expect that matters related to media composition and strain usage should be addressed more carefully and described more explicitly in the text, especially since nutritional complementation of auxotrophic strains is not always equivalent to genetic complementation [Pronk, 2002].

- A recent publication (Zeng et al. 2023, doi: 10.1016/j.celrep.2023.113599) identified Ser33 and Ser3 as TORC1 substrates and examined their dependence on Pib2 activity. More importantly, the publication addressed a question that is very similar to the one addressed here (i.e. how different amino acids require Gtr1/2 or Pib2 to activate TORC1). I would recommend that the authors cite that publication and compare their findings with the results reported there.

- The GO analysis of TORC1 substrates (from Fig.4) is mentioned in the text but is not shown. The authors should present the GO analysis more explicitly, e.g. in a supplementary table.

- Similar to Rps6, it should be kept in mind that Par32 is not a TORC1 substrate. While I understand the rationale behind the choice of Par32 as a readout, this point needs to be emphasized more. Additionally, previous work [Brito et al. 2019, doi: 10.1016/j.isci.2019.09.025] has suggested that Npr1 and Par32 are implicated in a feedback loop with Pib2. The potential relevance of that work should be discussed more here.

- Besides Sch9, Tod6 phosphorylation is also regulated by PKA [Huber et al. 2011, doi: 10.1038/emboj.2011.221]. This point should be discussed and taken into account in the interpretation of the Tod6 results. I also find it puzzling that Tod6 persists one hour after rapamycin treatment, because the protein seems to be unstable and gets quickly degraded when TORC1 activity is lost [Kusama 2022, doi: 10.1016/j.isci.2022.103986].

- Given the points raised above, I remain skeptical about the three-state model proposed by the authors. On a conceptual level, the intermediate activity state of TORC1 proposed here seems to depend absolutely on Pib2 (since Gtr1/2 appear to be off in that state). The authors make a similar point in the Discussion, where they claim that yeast growth on poor nitrogen sources can be halted by deletion of Pib2. However, they do not test this conjecture experimentally.

- Fig. 6F compares the growth of different strains on different media, but the doubling times are not quantified.

- The Introduction describes regulatory pathways of mTORC1, several of which do not exist in budding yeast. The transition from the second to third paragraph is very abrupt and confusing.

---

## [Referee Report · Reviewer #2 (Public Review)]

This work examines the roles of Gtr1/Gtr2 and Pib2 in activation of TORC1 in S cerevisiae and proposes they are non-redundant in activating TORC1. Previous work from many groups has suggested that the Gtr complex and Pib2 activate TORC1 in a parallel manner. One contribution of this study is the suggestion that using the standard readout(s) of TORC1 activation are not sufficient to assess the separate roles of these two components in the complex network of amino acid and starvation response signaling. The overall conclusion of the work, based on phosphoproteome analyses of deletion strains and comparison to rapamycin treatment, with some supporting experimentation, is that Pib2 signaling sustains the starvation response in poor amino acid/nitrogen sources, whereas the additional activation of the Gtr complex is required for the full spectrum of TORC1 effects on growth.

At first, the authors recapitulate and extend studies on TORC1 inactivation using the Rps6 reporter. Here, Pib2 could inactivate TORC1 on glutamine starvation only if the Gtr complex is partially compromised. The authors speculated that Gtr and Pib2 do lead to different responses, but these cannot be detected by monitoring the phospho state of Rps6.

The authors determined the phosphoproteome in wild type cells and a variety of knockout strains, in rich media and in the presence of rapamycin. The authors identified 175 phosphosites that are downregulated on rapamycin treatment, at least under these conditions. Many were dependent on both Pib2 and the Gtr complex but, of particular interest for this work , were the phosphosites on Ser33, that were dependent on the presence of Pib2 but not the Gtr complex. The authors noted that phosphosites not dependent on Pib2 or Gtr1/2 included Sch9 and other common readouts of TORC1 activation.

Focusing on Ser33, the authors next show that rapamycin, amino acid and nitrogen starvation result in loss of Ser33 phosphorylation. Further analysis showed that the Ser33 phosphorylation status depends on the quality of the amino acid and nitrogen source.

Then the authors use this to develop a model where TORC1 has three states depending on whether either Gtr1/2, or Pib2, or both are active in signaling to TORC1, depending on the nutrient state and quality of amino acids/nitrogen available. The new state is state III, where TORC1 is active to promote growth and the starvation response remains active, via the Npr1/Par32 branch. The remainder of the work involves developing tools to assess the growth (Sch9) and starvation (Par32) branches under various amino acid/nutrient states. While moving from media with an excess of all amino acids to glutamine or leucine led to only transient occupation of state III, the new state was already occupied when the cells were in a poor amino acid/nitrogen source and moved to a better one. In other words, the Pib2 signalling permitted aspects of a starvation response to be maintained in the background of a Sch9 growth signal.

Finally, the authors address a puzzle: Sch9 phosphorylation does not have the dynamic range to account for the difference in growth rates of yeast cells in SC or proline medium. Tod6 was dephosphorylated in the absence of Gtr1/Gtr2 or Pib2 in the phosphoproteomics and is the likely connection, as it moves to the nucleus on growth on proline media (or on rapamycin), where it may control the chromatin accessibility of ribosome growth and biogenesis genes.

Overall, the core of this work, the phosphoproteome analyses, convincingly demonstrates that activation of TORC1 relies on a nuanced interplay of signaling pathways and that to fully appreciate and dissect the consequences of the Gtr- and Pib2-responsive signaling pathways a more comprehensive range of readouts is required. The work elegantly shows a scenario where Pib2-based signaling is active, required to sustain some growth even when the amino acid/nitrogen mix is poor.

There are some areas, however, where the work could be strengthened. The model proposed in this work is based on nuanced signaling responses to various states of nitrogen/amino acid starvation. However, the phosphoproteome was determined in a synthetic rich background, supplemented with rapamycin where relevant, and comparing the phosphoproteome of pib2 del and gtr1 del/gtr2 del to this. The phosphoproteome is by far the strongest data in this work suggesting multi-level regulation so an appropriately matched phosphoproteome condition screen would likely significantly substantiate the model: the conditions used might miss all the nuanced signaling responses the authors develop throughout the paper. Not unrelated, the authors show that Pib2 can transmit glutamine starvation signals to TORC1 in the presence of a partial Gtr1/2 complex (gtr1 del or gtr2 del) but not a complete deletion of the complex (Fig. 2). Similar to the above comment, the phosphoproteome was determined only with full loss of the gtr complex, and then only in a rich background, which may miss this entire branch of Pib2 signaling. Perhaps in support of this, Pib2Ser113 phosphorylation apparently decreased significantly on rapamycin treatment but not on loss of the Gtr complex (TableS1), whereas other Pib2 phospho sites were not similarly affected by rapamycin treatment. Adding to the notion of complexity, the other sites may themselves be subject to other signaling pathways that could regulate Pib2 - and these may change on nutrient starvation.

The data showing the enrichment of Pib2 with Ser33 is weak (Fig. 5G, mostly because of the significant precipitation of Ser33 in the absence of Pib2), particularly without the contribution of the immunopurifications of Fig5S1. Assessing the binding of Ser3 may be a better candidate?

---

## [Referee Report · Reviewer #3 (Public Review)]

Summary:

This work addresses an important question of how Gtr1/2 small GTPases and Pib2, two major regulators of the TORC1 cell growth controller, differentially operate in yeast. They found not all the TORC1 downstream targets respond to Gtr1/2 and Pib2 equally. In fact, they demonstrate that TORC1-dependent phosphorylation of Ser33, a 3-phosphoglycerate dehydrogenase, is responsive to only Pib2. They attributed this specificity to the physical interaction between Ser33 and Pib2. This part is novel and important, revising the canonical view in the field that Gtr1/2 and Pib2 branches act towards the same TORC1 downstream targets. Of note, this claim largely agrees with a recent independent study (PMID: 38127619).

Moving on, the authors describe different behaviors of TORC1 downstream readouts in intermediate nutrient conditions with a poor nitrogen source, with some readouts still active while others inactive. They argue that selective activation of certain TORC1 downstream targets reflects the "Gtr1/2 off, Pib2 on" state. However, this claim is not sufficiently supported by the presented data.

Strengths:

The data presented in this paper has high value to the TOR community. In particular, a rigorous and comprehensive phospho-proteomic dataset that compares the Gtr1/2- and Pib2-dependency of diverse TORC1 downstream targets is very informative, potentially stimulating follow-up studies on each target.

Identification of Ser33 as a Pib2-specific TORC1 downstream is important and convincing (although whether Ser33 is a direct substrate of TORC1 was not addressed in this work). Physical interaction between Ser33 and Pib2 could represent a novel layer of TORC1 signaling regulation, in line with the mammalian Rag-TFEB interaction model, as discussed by the authors.

Weaknesses:

The authors' three-state model, particularly the claim that cells are in the "Gtr1/2 off, Pib2 on" state in a poor nitrogen condition (e.g., proline medium), is not convincing enough because of the following reasons.

1. The "Pib2 on" claim contradicts with the observation that Ser33, Pib2-specific readout, is hypo-phosphorylated in proline medium (Fig 5F).

2. In the genetic experiments (Figure 8), the authors compare pib2D with Gtr1/2OFF. This is not appropriate, because GTR1/2OFF (GTR1-GDP and Gtr2-GTP) actively inhibits TORC1, differing from the null nature of pib2D. pib2D should be compared with gtr1/2D instead.

3. In general, diverse behaviors of TORC1 targets are not unexpected because their phosphorylation levels should have different dynamic ranges depending on how "good" they are as TORC1 substrates, with some requiring a higher TORC1 activity than others to be detectably phosphorylated. Although this aspect can be physiologically meaningful, and it is indeed important to look at multiple substrates as the authors suggest, this approach does not inform whether the signal is coming from Gtr1/2 or Pib2. An informative way in this context would be to look at the Gtr1/2- or Pib2-specific targets, but the former has not been identified, and observations on the latter, Ser33, do not support the "Pib2 on" claim as mentioned in the above 1.

4. In addition, comparisons made between direct TORC1 substrates (e.g., Sch9) and indirect downstream targets (e.g., Rps6 and Par32) are not very informative, because indirect targets can be impacted by TORC1-independent regulation of the mediating factors (e.g., Ypk3 for Rps6 and Npr1 for Par32).

In summary, the presented data do not tell us which of the two branches (Gtr1/2 or Pib2) is "more active" in the poor nitrogen condition. Their observations do not necessarily prefer their 3-state on/off model (Figure 8) over the more natural assumption that both branches have the gradation of activity depending on the nutrient status.

---

## [Author Response]

We thank the reviewers for their thoughtful and constructive feedback. As the reviewers noted, dissecting the contributions of Gtr1/2 and Pib2 to TORC1 signaling across diverse nutrient states is a technically and conceptually challenging problem. Indeed, many of the issues raised—including the interpretation of non-canonical TORC1 readouts (e.g., Rps6, Par32), the influence of strain auxotrophy and media composition, and the limitations of phosphoproteomic analysis performed under a single growth condition—underscore the challenges of working with the TORC1 signaling system.

In response to the reviewers’ comments, we have undertaken a broader and more systematic analysis of TORC1 regulation across defined nitrogen transitions, building directly on the signaling framework established in Figures 6 and 8 of this manuscript. This work, which includes expanded phosphoproteomic profiling and the use of refined genetic tools, supports and extends the key conclusions of Cecil et. al. Specifically, it reinforces the existence of a Pib2-dependent TORC1 output under nitrogen-limited conditions and further clarifies the physiological relevance of the intermediate TORC1 activity state. Due to the scope and depth of this expanded work, we are reporting those findings in a separate publication. Nonetheless, we view the data presented here as a key foundational step in establishing a non-redundant framework for Gtr1/2- and Pib2-dependent control of TORC1.

We have therefore made minor changes to the manuscript to clarify our use of different growth media and to temper our conclusions where appropriate. These changes, together with the context of ongoing work, should reinforce the value of Cecil et. al. in advancing our understanding of TORC1 and nutrient signaling in eukaryotes.